# The universal suppressor mutation restores membrane budding defects in the HSV-1 nuclear egress complex by stabilizing the oligomeric lattice

Elizabeth B. Draganova[1¤], Hui Wang[2,3,4], Melanie Wu[5], Shiqing Liao[2,4], Amber Vu[6], Gonzalo L. Gonzalez-Del Pino[1], Z. Hong Zhou[2,3,4], Richard J. Roller[6], Ekaterina E. Heldwein[1]*

1 Department of Molecular Biology and Microbiology, Tufts University School of Medicine, Boston, Massachusetts, United States of America, 2 Department of Microbiology, Immunology & Molecular Genetics, University of California, Los Angeles (UCLA), Los Angeles, California, United States of America, 3 Department of Bioengineering, UCLA, Los Angeles, California, United States of America, 4 California NanoSystems Institute, UCLA, Los Angeles, California, United States of America, 5 School of Chemistry and Molecular Biosciences, The University of Queensland, Brisbane, Australia, 6 Department of Microbiology and Immunology, Carver College of Medicine, University of Iowa, Iowa City, Iowa, United States of America

¤ Current address: Department of Biochemistry, Emory University School of Medicine, Atlanta, Georgia, United States of America
* katya.heldwein@tufts.edu

## Abstract

Nuclear egress is an essential process in herpesvirus replication whereby nascent capsids translocate from the nucleus to the cytoplasm. This initial step of nuclear egress–budding at the inner nuclear membrane–is coordinated by the nuclear egress complex (NEC). Composed of the viral proteins UL31 and UL34, NEC deforms the membrane around the capsid as the latter buds into the perinuclear space. NEC oligomerization into a hexagonal membrane-bound lattice is essential for budding because NEC mutants designed to perturb lattice interfaces reduce its budding ability. Previously, we identified an NEC suppressor mutation capable of restoring budding to a mutant with a weakened hexagonal lattice. Using an established in-vitro budding assay and HSV-1 infected cell experiments, we show that the suppressor mutation can restore budding to a broad range of budding-deficient NEC mutants thereby acting as a universal suppressor. Cryogenic electron tomography of the suppressor NEC mutant lattice revealed a hexagonal lattice reminiscent of wild-type NEC lattice instead of an alternative lattice. Further investigation using x-ray crystallography showed that the suppressor mutation promoted the formation of new contacts between the NEC hexamers that, ostensibly, stabilized the hexagonal lattice. This stabilization strategy is powerful enough to override the otherwise deleterious effects of mutations that destabilize the NEC lattice by different mechanisms, resulting in a functional NEC hexagonal lattice and restoration of membrane budding.

**Data Availability Statement:** All data generated or analyzed during this study are included in the manuscript and supporting files. A source data file

is provided for data presented in Figs 1–4. The cryo-ET sub-tomogram average maps have been deposited in the EM Data Bank under the accession codes EMD-40223 (WT NEC), EMD-40224 (NEC-DNUL34/SUPUL31), and EMD-40225 (NEC-SUPUL31). Atomic coordinates and structure factors for the NEC-SUPUL31 crystal structure have been deposited in the RCSB Protein Data Bank under accession code 8G6D.

**Funding:** EEH was supported by grants R01GM111795 and R01AI147625 from the National Institutes of Health and by a Faculty Scholar grant 55108533 from Howard Hughes Medical Institute. ZHZ was supported by grant R01DE028583 from the National Institutes of Health. RJR was supported by grant R01AI150718 and R21AI148831 from the National Institutes of Health. EBD was supported by grants F32GM126760, K12GM133314, and K99AI151891 from the National Institutes of Health. The funders had no role in study design, data collection and analysis, decision to publish, or preparation of the manuscript.

**Competing interests:** Authors declare no competing interests.

## Author summary

Herpesviruses infect most of the human population, establishing lifelong infections that periodically reactivate. Successful infection of a host requires the formation of new viral particles within host cells. Early in viral assembly, capsids are translocated from the nucleus into the cytoplasm. This process of capsid nuclear egress is mediated by the virally encoded nuclear egress complex (NEC) that deforms the membrane around the capsid as it oligomerizes into a hexagonal lattice. Mutations that disrupt the lattice can block nuclear egress and reduce viral progeny production, making the NEC an attractive therapeutic target. However, the virus can overcome the inhibitory effects of such mutations through suppressor mutations. Here, we show that one such suppressor mutation promotes new lattice contacts that are expected to strengthen the lattice. We hypothesize that the NEC can fine-tune its lattice in response to external perturbations, be they natural such as membrane curvature, or artificial such as mutations. Any therapeutic approach that targets NEC lattice formation must consider its intrinsic flexibility.

## Introduction

Viruses are experts at reorganizing host membranes to traffic their capsids across the compartmentalized interior of eukaryotic cells. One of the more unusual mechanisms of membrane manipulation is found in *Herpesvirales*, which is an order of large, enveloped viruses that infect multiple species across the animal kingdom and cause life-long infections in the majority of the world's population [reviewed in [1]]. Replication of herpesviral dsDNA genomes and their subsequent packaging into capsids occurs within the nucleus [reviewed in [2,3]]. Genome-containing capsids are then transported into the cytoplasm for maturation into infectious virions. Most of the nucleocytoplasmic traffic occurs through the nuclear pores, but at ~125 nm in diameter, herpesviral capsids are too large to fit through the ~40-nm nuclear pore opening [4]. Instead, the capsids use a complex, non-canonical nuclear transport route termed nuclear egress [reviewed in [2,3,5]]. First, they dock at the inner nuclear membrane (INM) and bud into the perinuclear space, producing perinuclear enveloped virions (a stage termed primary envelopment). The envelopes of these intermediates then fuse with the outer nuclear membrane (ONM) and capsids are then released into the cytoplasm (a stage termed de-envelopment).

The nuclear egress mechanism is best understood for the family *Herpesviridae*, commonly referred to as herpesviruses, which infect mammals, birds, and reptiles. Two conserved viral proteins, called UL31 and UL34 in herpes simplex virus (HSV), are essential for nuclear egress in herpesviruses. UL31 is a soluble nuclear phosphoprotein [6, 7] whereas UL34 is a type I membrane protein containing a single C-terminal transmembrane helix [6, 8]. Together, UL31 and UL34 form the heterodimeric nuclear egress complex (NEC) that is anchored in the INM and faces the nuclear interior. Both proteins are essential for nuclear egress, and in the absence of either, capsids become trapped within the nucleus, which results in greatly reduced viral titers [6,7,9–13]. Moreover, overexpression of both UL31 and UL34 in uninfected cells causes the accumulation of empty budded vesicles in the perinuclear space, which implies that the NEC is not only necessary but also sufficient for the INM budding [14–18]. Collectively, these findings highlight the central role of the NEC during nuclear egress.

Recent studies with purified recombinant NEC and synthetic lipid vesicles have shown that several NEC homologs can deform and bud membranes *in vitro* in the absence of added energy or other proteins. These include the NECs from herpes simplex virus 1 (HSV-1) [19], a

prototypical herpesvirus that infects much of the world's population; the closely related pseudorabies virus (PRV) that infects animals [20]; and the more distantly related Epstein-Barr Virus (EBV) [21], a nearly ubiquitous human herpesvirus.

The NEC oligomerizes into membrane-bound coats on the inner surface of the budded vesicles. Hexagonal coats resembling a honeycomb have been observed by cryogenic electron microscopy/tomography (cryoEM/ET) on vesicles formed by recombinant HSV-1 NEC *in vitro* [19], vesicles formed in uninfected cells overexpressing PRV NEC [22], and in perinuclear vesicles formed in HSV-1-infected cells [23]. Interestingly, crystallized NEC homologs from HSV-1 [24] and human cytomegalovirus (HCMV) [25] also formed hexagonal crystal lattices of geometry and dimensions similar to those observed in the membrane-bound NEC coats. Finally, EBV NEC also forms membrane-bound coats *in vitro* but their geometry is yet unclear [21]. Both the intrinsic membrane budding ability and the formation of oligomeric coats thus appear to be conserved among the NEC homologs.

In HSV-1 NEC, oligomerization into the hexagonal lattice is essential for budding. Mutations targeting lattice interfaces within the NEC hexamers (hexameric) or between hexamers (interhexameric) cause budding defects *in vitro* [19,24] and reduce nuclear egress in infected cells [26,27]. The first such mutation, which changed D35 and E37 in UL34 to alanine (A), was identified in a mutational screen targeting charge clusters in the HSV-1 UL34 sequence [28]. Recognizing that the mutations are in the gene rather than the protein, for clarity, we will refer to the proteins translated from the mutant genes as mutant proteins designated by the altered amino acids (in this case, as the $D35A_{UL34}/E37A_{UL34}$ mutant). This double mutation reduced viral titers by ~3 orders of magnitude to levels of UL34-null mutant HSV-1 and blocked capsid egress from the nucleus in a dominant-negative manner [27]. Therefore, we refer to it as $DN_{UL34}$. The mutation did not affect the NEC formation, its localization to the INM, or capsid docking at the INM but, instead, precluded capsid budding [27]. Furthermore, purified recombinant NEC-$DN_{UL34}$ bound synthetic membranes *in vitro* but had minimal membrane-budding activity and did not form hexagonal coats on membranes [19].

Interestingly, the nuclear budding defect due to the $DN_{UL34}$ mutation could be suppressed by an extragenic mutation in HSV-1 UL31, $R229L_{UL31}$, which arose during serial passaging of the $DN_{UL34}$ mutant HSV-1 virus on a UL34-complementing cell line [27] through directed viral evolution [reviewed in [29,30]]. We refer to this mutation as $SUP_{UL31}$. Residue $R229_{UL31}$ is located near the interhexameric interface, far away from the $DN_{UL34}$ mutations at the hexameric interface [24], making it unclear how the $SUP_{UL31}$ mutation restores $DN_{UL34}$ nuclear budding defects.

Here, we show that the $SUP_{UL31}$ mutation can restore efficient budding to a broad range of mutants that disrupt important functional interfaces, acting as a "universal" suppressor of budding defects. Using cryogenic electron tomography (cryoET) and x-ray crystallography, we show that the $SUP_{UL31}$ mutation does not change the structure of the NEC heterodimer or its oligomerization into hexamers. Instead, it promotes the formation of new contacts at the interhexameric interface. The increased interhexameric interface would reinforce the hexagonal NEC lattice, helping it to counteract the lattice-destabilizing effects of mutations. We hypothesize that its dynamic nature allows the NEC lattice to adapt to perturbations that it encounters during nuclear egress, for example, changes in membrane curvature or capsid interactions, and maintain its function.

## Results

### The $SUP_{UL31}$ mutation restores membrane budding *in vitro* to various oligomeric interface mutants

HSV-1 NEC oligomerizes into a hexagonal lattice (**Fig 1A**) stabilized by interactions between NEC heterodimers within hexamers (hexameric interface; **Fig 1B**) and between hexamers

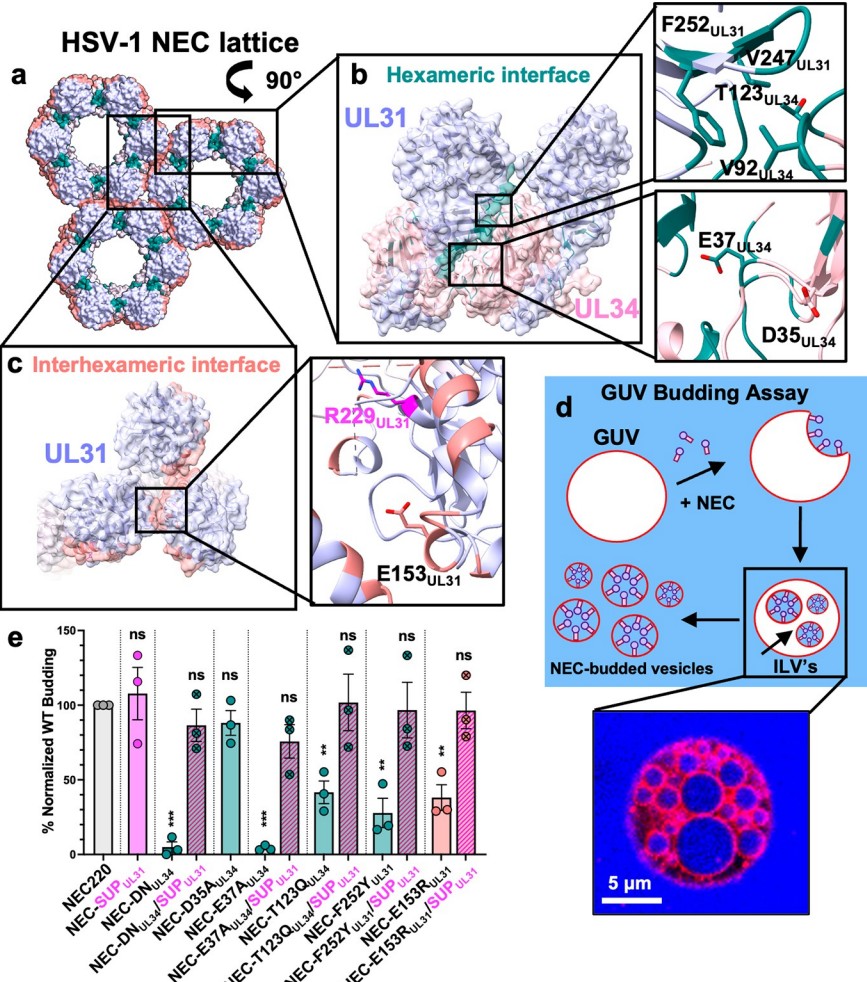

**Fig 1. The SUP_{UL31} mutation restores budding activity to budding-deficient oligomeric interface NEC mutants.**
**a-c)** HSV-1 NEC hexameric and interhexameric interfaces highlighting the locations of residues mutated for this study. **d)** A cartoon representation of the GUV budding assay showing the NEC (purple circles and pink rectangles) binding to red fluorescent GUVs and undergoing negative curvature to form an NEC-coated intraluminal vesicle (ILV). Free NEC continues to bud the GUVs until only fully budded vesicles containing NEC on the interior remain. Cascade blue, a membrane impermeant dye, is used to monitor budding. **e)** SUP-R229L_{UL31} rescues budding in both hexameric and interhexameric budding-deficient NEC mutants *in vitro*. The percentage of *in vitro* budding was determined by counting the number of ILVs within the GUVs after the addition of NEC220 or the corresponding NEC mutant. A background count, the number of ILVs in the absence of NEC, was subtracted from each condition. Each construct was tested in three biological replicates, each consisting of three technical replicates. Symbols show the average budding efficiency of three technical replicates compared to NEC220 (100%; grey). Significance was calculated using an unpaired Student's t-test with Welch's correction (P < 0.01 = **; P < 0.001 = ***; ns = not significant) in GraphPad Prism 9.0.

(interhexameric interface; **Fig 1C**). NEC hexameric lattice formation is essential for membrane budding because mutations engineered to disrupt lattice interfaces reduce budding *in vitro* [19,24] and nuclear egress in infected cells [26,27]. These budding-deficient mutations are D35A_{UL34}/E37A_{UL34} (DN_{UL34}), V92F_{UL34}, T123Q_{UL34}, V247F_{UL31}, and F252Y_{UL31} at the hexameric interface (**Fig 1B**) and E153R_{UL31} at the interhexameric interface (**Fig 1C**). The SUP_{UL31} mutation restores budding *in vitro* to DN_{UL34} and V92F_{UL34} mutants [24]. Here, we asked if it could restore budding to other interface mutants, T123Q_{UL34} and F252Y_{UL31} (hexameric) (**Fig 1B**) and E153R_{UL31} (interhexameric) (**Fig 1C**).

We also wanted to assess the individual contributions of $D35A_{UL34}$ and $E37A_{UL34}$ mutations to the budding-deficient phenotype of the $DN_{UL34}$ mutant. Residue $E37_{UL34}$ is located at the hexameric interface where its side chain forms a hydrogen bond with $T89_{UL31}$ of the neighboring NEC heterodimer. The $E37A_{UL34}$ mutation eliminates this hydrogen bond, which would disrupt the hexameric interface. Indeed, the $E37A_{UL34}$ mutant was deficient in budding *in vitro* [24]. However, the side chain of residue $D35_{UL34}$ points away from the hexameric interface (**Fig 1B**). Therefore, we tested if the $D35A_{UL34}$ mutation would have any effect on budding.

The *in-vitro* budding activity of all NEC mutants was measured by an established assay [19,24,31,32] utilizing fluorescently labeled giant unilamellar vesicles (GUV), soluble fluorescent dye Cascade Blue, and the soluble version of HSV-1 NEC, NEC220, which contained fulllength UL31 and UL34 residues 1–220 (**Fig 1D**). We first confirmed the *in-vitro* phenotypes of the budding-deficient mutants. Both $DN_{UL34}$ and $E37A_{UL34}$ mutations reduced budding to ~10% of the WT NEC220 (**Fig 1E**), consistent with our previous findings [24] whereas the $D35A_{UL34}$ mutation alone had no effect (**Fig 1E**). Thus, the $E37A_{UL34}$ mutation is solely responsible for the nonbudding phenotype of $DN_{UL34}$. The interface mutations $T123Q_{UL34}$, $F252Y_{UL31}$, and $E153R_{UL31}$ (**Fig 1BC**) reduced budding to ~30–40% of the WT NEC220 (**Fig 1E**), as previously observed [24]. The $SUP_{UL31}$ mutation did not affect the budding efficiency of the WT NEC220 but restored budding not only to the $DN_{UL34}$ as shown previously [24] but to all other lattice interface mutants regardless of their location (**Fig 1E**). Thus, the $SUP_{UL31}$ mutation can restore efficient budding to a broad range of lattice interface mutants.

## The $SUP_{UL31}$ mutation complements the growth defects of HSV-1 containing oligomeric interface mutations

To correlate the *in-vitro* budding phenotypes with the infected cell phenotypes, we used an established viral growth complementation assay [11]. This assay measures the ability of a mutant protein expressed *in trans* to complement the poor growth of a virus lacking the corresponding gene (the so-called null virus). Hep-2 cells were transfected with plasmids encoding either WT (UL31 or UL34), mutant UL34 ($D35A_{UL34}/E37A_{UL34}$, $D35A_{UL34}$, $E37A_{UL34}$, $T123Q_{UL34}$), or mutant UL31 ($F252Y_{UL31}$ and $E153R_{UL31}$) and then infected with either a UL34-null HSV-1 or UL31-null HSV-1. The amount of infectious viral progeny produced was measured by plaque assay on either UL34-expressing (**Fig 2A**) or UL31-expressing Vero cells (**Fig 2B**). We found that cells expressing the $D35A_{UL34}/E37A_{UL34}$, $E37A_{UL34}$, $T123Q_{UL34}$, or $E153R_{UL31}$ mutants did not trans-complement replication of either the UL34-null (**Fig 2A**) or UL31-null HSV-1 (**Fig 2B**), as efficiently as the WT UL34 or WT UL31, respectively. This is in agreement with their impaired budding phenotypes *in vitro*. Although not statistically significant, the reduced trans-complementation ability of $E37A_{UL34}$ and $T123Q_{UL34}$ was evident. By contrast, the $D35A_{UL34}$ mutant trans-complemented UL34-null HSV-1 with WT UL34 efficiency (**Fig 2A**). Surprisingly, the $F252Y_{UL31}$ mutant trans-complemented UL31-null HSV-1 with the WT UL31 efficiency despite reduced budding in our *in-vitro* budding assay (**Fig 1E**).

To rule out increased protein expression levels as the trans-complementation mechanism, we measured expression levels of transfected WT UL34, WT UL31, and the corresponding mutant proteins during infection with the corresponding null virus. All mutant UL34 proteins expressed at WT UL34 levels (**S1A Fig**). Reduced complementation efficiencies of $D35A_{UL34}/E37A_{UL34}$, $E37A_{UL34}$, and $T123Q_{UL34}$ (**Fig 2A**) are thus due to the specific mutation(s). In contrast, both $F252Y_{UL31}$ and $E153R_{UL31}$ are overexpressed relative to the WT UL31 and $R229L_{UL31}$ (**S1B Fig**). The surprisingly efficient complementation by the $F252Y_{UL31}$ mutant may therefore be due to its higher expression levels (**Fig 2B**). We have not, however, generated

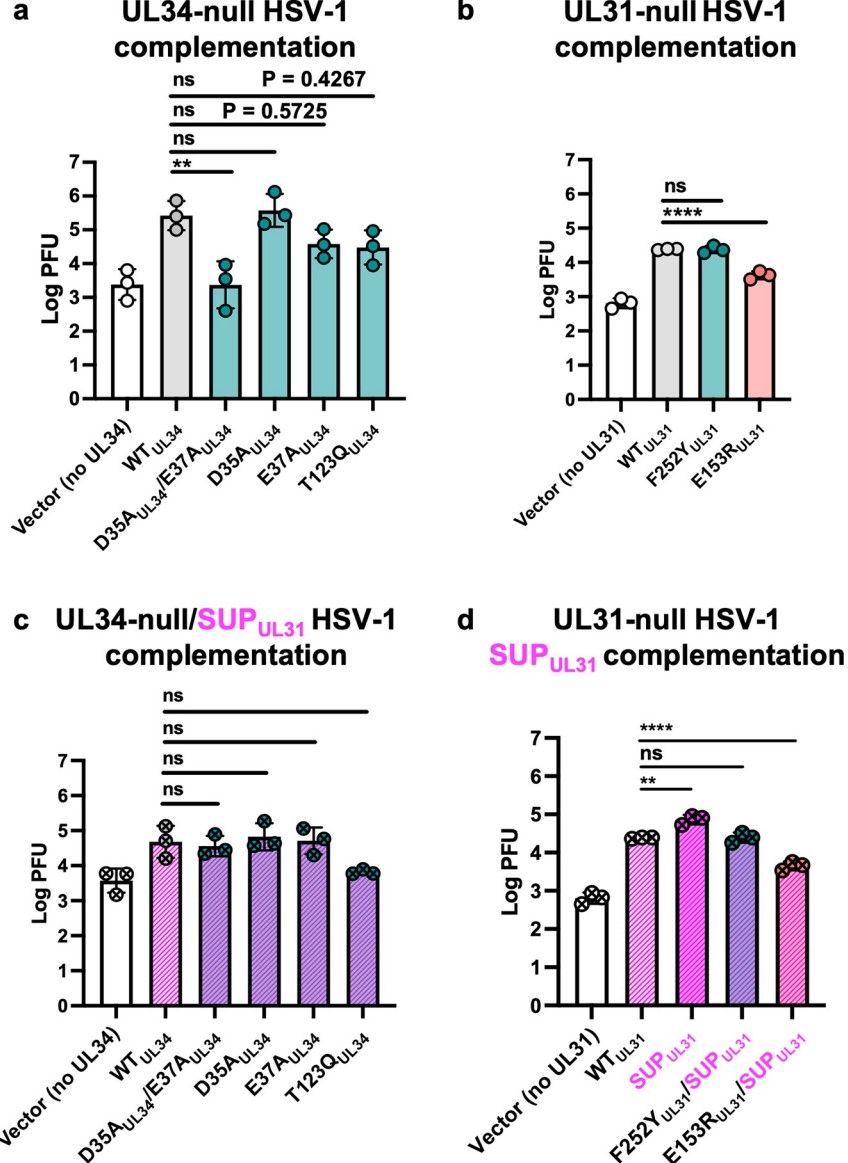

**Fig 2. The SUP$_{UL31}$ mutation complements the growth defects of HSV-1 containing oligomeric interface mutations. (a-b)** WT$_{UL31}$ can only complement the growth of the D35A$_{UL34}$ and F252Y$_{UL31}$ oligomeric interface mutants whereas SUP$_{UL31}$ can complement more **(c-d).** For all experiments, Hep-2 cells were transfected with the corresponding UL34 or UL31 mutant plasmid (x-axis labels) and infected with either a UL34-null **(a)**, a UL31-null virus **(b)**, a UL34-null/UL31$_{R229L}$ virus **(c)**, or a UL31-null virus with the SUP$_{UL31}$ mutation added in *trans* **(d)**. Each bar represents the mean of three independent experiments. Statistical significance was determined by performing a one-way ANOVA on log-converted values using the Method of Tukey for multiple comparisons of each mutant against the WT in GraphPad Prism. P < 0.01 = **; P < 0.0001 = ****; ns, not significant.

a recombinant virus expressing F252Y$_{UL31}$ to assess its function at physiological expression levels to confirm this.

Next, we tested the ability of the SUP$_{UL31}$ mutation to restore efficient complementation to the lattice interface mutants, i.e., to suppress their complementation defects. To test the UL34 mutants, we generated the UL34-null/SUP$_{UL31}$ HSV-1 virus. Hep-2 cells were transfected with plasmids encoding either WT UL34 or mutant UL34 (D35A$_{UL34}$/E37A$_{UL34}$, D35A$_{UL34}$,

E37A$_{UL34}$, or T123Q$_{UL34}$) and then infected with a UL34-null/SUP$_{UL31}$ HSV-1. UL34-null/ SUP$_{UL31}$ HSV-1 yields ~1-log-fold lower viral titer when complemented with WT UL34 (**Fig 2C**) than the UL34-null HSV-1 complemented with WT UL34 (**Fig 2A**), as reported previously [27]. Therefore, complementation of the UL34-null/SUP$_{UL31}$ virus by the WT UL34 was used as a reference point for assessing the ability of the SUP$_{UL31}$ mutation to restore efficient complementation to UL34 mutants. Indeed, the SUP$_{UL31}$ mutation rescued complementation defects of both the D35A$_{UL34}$/E37A$_{UL34}$ and E37A$_{UL34}$ mutants (**Fig 2C**), both of which poorly complemented the growth of the UL34-null HSV-1 (**Fig 2A**). The SUP$_{UL31}$ mutation had no obvious effect on the already efficient complementation by the D35A$_{UL34}$ (**Fig 2C**). Unexpectedly, it did not fully restore the poor complementation by the T123Q$_{UL34}$ (**Fig 2C**) despite restoring its budding defect *in vitro* (**Fig 1E**). We note, however, that the difference in complementation levels between the T123Q$_{UL34}$ mutant and the WT UL34 are not statistically significant (**Fig 2C**).

To test the UL31 mutants, we generated double UL31 mutant plasmids harboring the UL31 mutation of interest and the SUP$_{UL31}$ mutation. Hep-2 cells were transfected with plasmids encoding either WT UL31 or mutant UL31 (R229L$_{UL31}$, F252Y$_{UL31}$/R229L$_{UL31}$, or E153R$_{UL31}$/ R229L$_{UL31}$, and then infected with the UL31-null HSV-1. Complementation of UL31-null viruses by the WT UL31 was used as a reference point for assessing the ability of the SUP$_{UL31}$ mutation to restore efficient complementation to UL31 mutants. The SUP$_{UL31}$ mutation complemented the growth defect of the UL31-null HSV-1 somewhat more efficiently than the WT UL31. We also occasionally observe more efficient budding by the NEC-SUP$_{UL31}$ in our *in vitro* budding assays [24] but the reason for this is unclear. In HSV-1 UL31-null infected cells, the SUP$_{UL31}$ mutation had no obvious effect on the already efficient complementation by the F252Y$_{UL31}$ mutant (**Fig 2D**). However, it was unable to fully restore the poor complementation by the E153R$_{UL31}$ mutant (**Fig 2D**) despite restoring its budding defect *in vitro* (**Fig 1E**).

The reason for poor complementation of the T123Q$_{UL34}$ and E153R$_{UL31}$ mutations by the SUP$_{UL31}$ mutation is yet unclear. We hypothesize that these mutations may impair some other important viral replication function of UL34 or UL31, respectively, that cannot be suppressed by the SUP$_{UL31}$ mutation, e.g., nuclear lamina dissolution, capsid docking at the INM, or capsid recruitment.

## The SUP$_{UL31}$ mutation restores efficient budding *in vitro* to heterodimeric interface mutants and complements their viral growth defects

The aforementioned mutational screen targeting charge clusters in the HSV-1 UL34 sequence [28], identified another double mutant, K137A$_{UL34}$/R139A$_{UL34}$, that could not trans-complement the growth of the HSV-1 UL34-null virus. This suggested that residues K137$_{UL34}$ and R139$_{UL34}$ are important for HSV-1 replication. The double mutation did not affect the NEC localization to the INM, suggesting a defect in the NEC function [28]. In the HSV-1 NEC crystal structure, K137$_{UL34}$ forms salt bridges with E67$_{UL34}$ and Y61$_{UL31}$ at the heterodimeric interface between the globular domains of UL31 and UL34 (**Fig 3A, inset**). Thus, K137$_{UL34}$ could contribute to the stabilization of the NEC heterodimer. By contrast, R139$_{UL34}$ does not form any obvious interactions (**Fig 3A, inset**).

To test the effect of the K137A$_{UL34}$, R139A$_{UL34}$, and K137A$_{UL34}$/R139A$_{UL34}$ mutations on the heterodimer stability and budding activity *in vitro*, we introduced them into the recombinant NEC220. Typically, size-exclusion chromatography on samples of purified, WT NEC220 yields only fractions containing equimolar amounts of UL31 and UL34, indicating the intact UL31:UL34 = 1:1 complex [19]. Indeed, this pattern was observed for the NEC220-R139A$_{UL34}$ mutant (**S2A Fig**). However, both NEC220-K137A$_{UL34}$ and NEC220-K137A$_{UL34}$/R139A$_{UL34}$

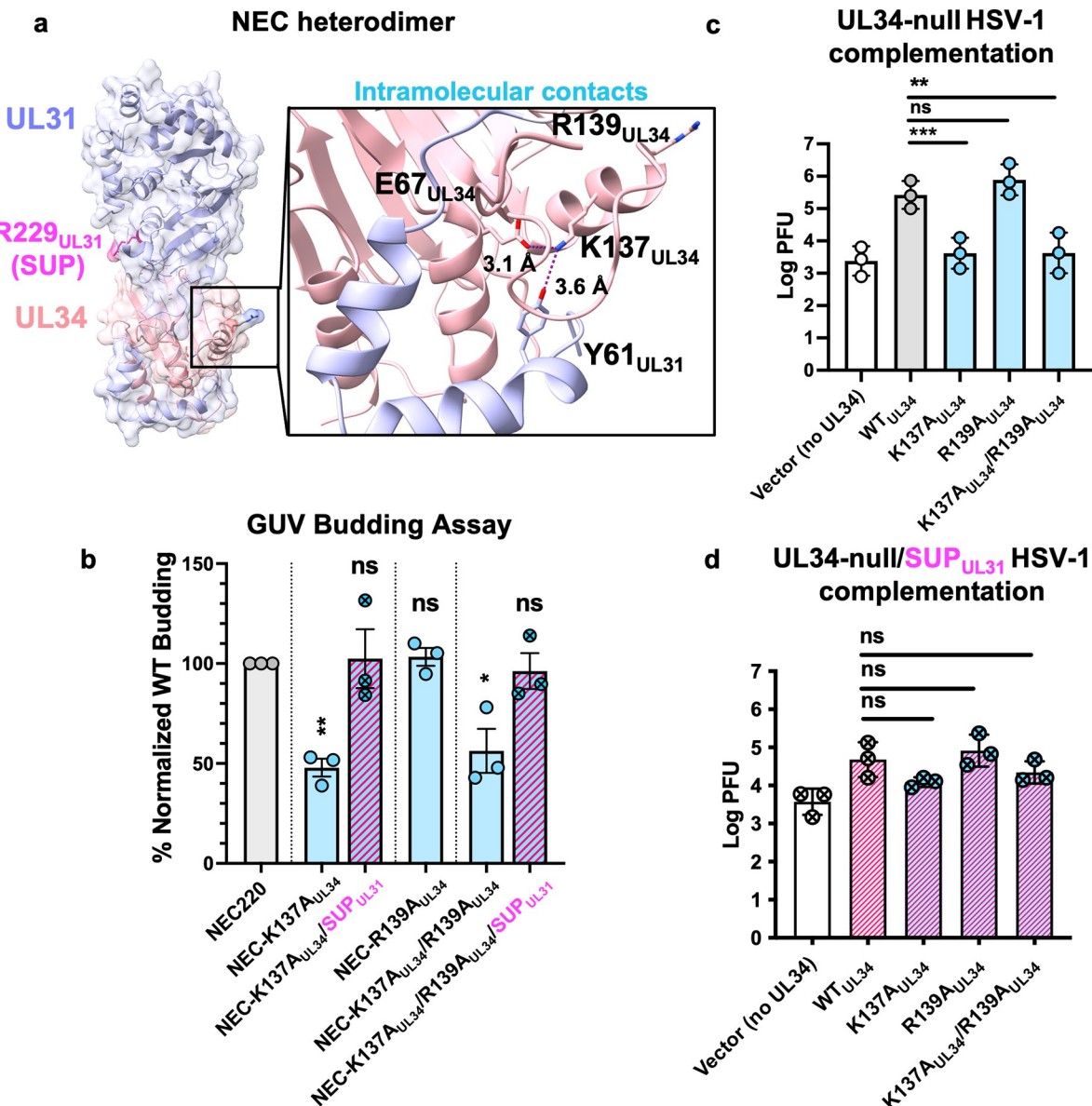

**Fig 3. The SUP_UL31 mutation restores budding to heterodimeric interface mutants and complements viral growth defects. a)** Locations of intramolecular NEC residues mutated for this study. Inset shows interactions between various residues at the heterodimeric interface thought to be important for NEC heterodimer stabilization. **b)** NEC-SUP_UL31 rescues budding of NEC heterodimeric interface mutants *in-vitro*. The percentage of *in-vitro* budding was determined by counting the number of ILVs within the GUVs after the addition of NEC220 or the corresponding NEC mutant. A background count, the number of ILVs in the absence of NEC, was subtracted from each condition. Each construct was tested in three biological replicates, each consisting of three technical replicates. Symbols show the average budding efficiency of three technical replicates compared to NEC220 (100%; grey). Significance was calculated using an unpaired Student's t-test with Welch's correction (P < 0.05 = *; P < 0.01 = **; ns = not significant) in GraphPad Prism 9.0. **c)** WT_UL34 can only complement the growth of the R139A_UL34 heterodimeric interface mutant whereas SUP_UL31 **(d)** can partially complement more. For both experiments, Hep-2 cells were transfected with the corresponding UL34 mutant plasmid and infected with a UL34-null virus **(c)** or a UL34-null/UL31_R229L virus **(d)**. Each bar represents the mean of three independent experiments. Statistical significance was determined by performing a one-way ANOVA on log-converted values using the Method of Tukey for multiple comparisons implemented on GraphPad Prism. **, P < 0.01 = **; P < 0.001 = ***; ns, not significant.

mutants also yielded fractions containing free UL34 or fractions containing more UL34 than UL31 (**S2B and S2C Fig**) despite equimolar amounts of UL31 and UL34 being loaded onto the size-exclusion column. Thus, the $K137A_{UL34}$ mutation appeared to destabilize the NEC heterodimer. No free UL31 was detected in any of the fractions, suggesting that it may have been retained on a filter within the chromatography line. Only fractions containing equimolar amounts of UL31 and UL34 were used for further characterization.

Both $K137A_{UL34}$ and $K137A_{UL34}$/$R139A_{UL34}$ mutations reduced budding to ~50% of the WT NEC220 whereas the $R139A_{UL34}$ mutation had no effect (**Fig 3B**). The $K137A_{UL34}$ mutation is thus solely responsible for the defective budding phenotype of the double $K137A_{UL34}$/$R139A_{UL34}$ mutant. Surprisingly, the $SUP_{UL31}$ mutation fully restored efficient budding to both $K137A_{UL34}$ and $K137A_{UL34}$/$R139A_{UL34}$ mutants (**Fig 3B**). But the mutant NEC heterodimers still eluted as fractions containing either free UL34 or mutant NEC complex, suggesting the mutant NEC complexes remained unstable despite the presence of the $SUP_{UL31}$ mutation (**S3 Fig**). Therefore, the $SUP_{UL31}$ mutation does not restore budding by restoring heterodimer stability.

To assess the effects of these mutations on viral replication, we performed the viral growth complementation assay described above for UL34 mutants. Both $K137A_{UL34}$ and $R139A_{UL34}$ proteins were expressed at WT UL34 levels (**S1A Fig**). $R139A_{UL34}$ complemented the growth of both the UL34-null (**Fig 3C**) and UL34-null/$SUP_{UL31}$ viruses on par with the WT UL34 (**Fig 3D**). As expected, $K137A_{UL34}$ and $K137A_{UL34}$/$R139A_{UL34}$ complemented growth of the UL34-null virus poorly (**Fig 3C**), which is consistent with their *in-vitro* budding defects. However, both mutants complemented the growth of the UL34-null/$SUP_{UL31}$ virus almost as efficiently as the WT UL34 (**Fig 3D**). Therefore, $SUP_{UL31}$ mutation can restore both budding and replication defects caused by the $K137A_{UL34}$ mutation.

## The $SUP_{UL31}$ mutation partially restores budding *in vitro* to a membrane interface mutant

In addition to UL31/UL34 and NEC/NEC interfaces, the NEC/membrane interface is also functionally important in HSV-1 NEC. Both UL31 and UL34 contain membrane-proximal regions (MPRs) (**Fig 4A and 4B**) that mediate membrane association [19,31] and are essential for budding *in vitro* [31]. The UL31 MPR contains clusters of positively charged residues that interact with model membranes and facilitate membrane deformation and budding by peripherally inserting into the membrane and increasing lipid order [31]. The UL31 MPR also contains six serines (**Fig 4B**) that are phosphorylated during infection [7] by the HSV-1 kinase US3 [33]. Phosphomimicking serine-to-glutamate mutations of these six serines ($SE6_{UL31}$) (**Fig 4B**) reduce nuclear egress and viral titers during HSV-1 infection [34] and impair NEC/ membrane interactions and budding activity *in vitro* [31]. Previously, we proposed that negative charges introduced by phosphorylation or phosphomimicking mutations reduce electrostatic interactions between the MPR and the lipid headgroups that are necessary for membrane deformation and budding [31]. Here, we asked whether the $SUP_{UL31}$ mutation could restore budding *in vitro* to the budding deficient $SE6_{UL31}$ mutant.

The NEC220 construct typically used in the *in-vitro* budding assays is soluble and depends on functional MPRs for membrane recruitment. Since the $SE6_{UL31}$ mutation reduces NEC/ membrane interactions, to bypass the defect in membrane recruitment, we used the NEC220 variant construct that contains a $His_8$-tag at the C terminus of UL34 [19,31]. When used in conjunction with membranes containing Ni-chelating lipids, the $His_8$-tag ensures that the NEC220-$His_8$ is recruited to the membranes even when the MPR mutations preclude membrane association. The *in-vitro* budding efficiency of NEC220-$His_8$ was similar to that of

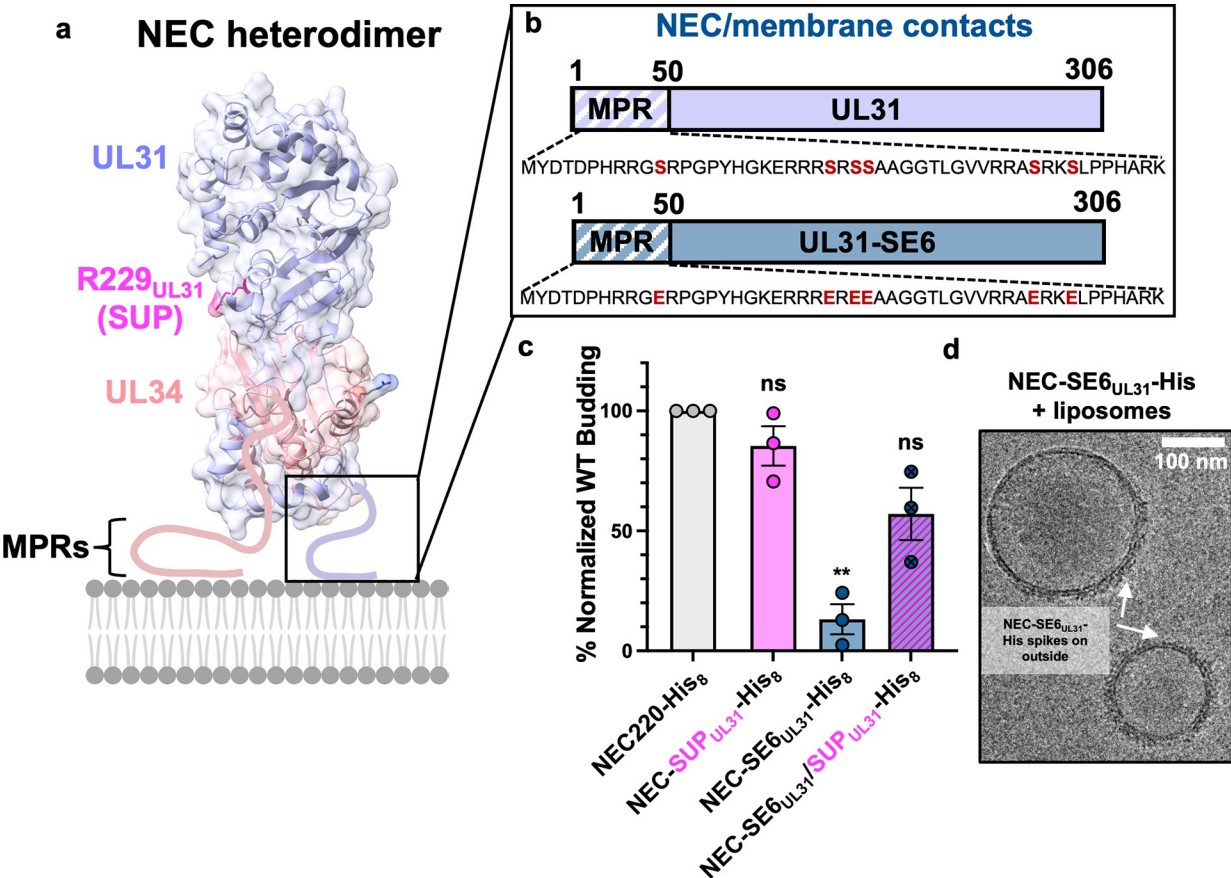

**Fig 4. The SUP$_{UL31}$ mutation partially restores budding to a membrane interface mutant. a-b)** Location of membrane interface residues mutated for this study. **c)** NEC-SUP$_{UL31}$ partially rescues budding in the membrane interface mutant *in-vitro*. The percentage of *in-vitro* budding was determined by counting the number of ILVs within the GUVs after the addition of NEC220-His or the corresponding NEC mutant. A background count, the number of ILVs in the absence of NEC, was subtracted from each condition. Each construct was tested in three biological replicates, each consisting of three technical replicates. Symbols show the average budding efficiency of three technical replicates compared to NEC220-His$_8$ (100%; grey). The NEC-SE6$_{UL31}$-His$_8$ data were previously reported in [31]. Significance was calculated using an unpaired Student's t-test with Welch's correction (P < 0.01 = **; ns = not significant) in GraphPad Prism 9.0. **d)** CryoEM of NEC-SE6$_{UL31}$-His$_8$ and large unilamellar vesicles (LUVs) shows that the SE6$_{UL31}$ mutations perturb NEC oligomerization when bound to membranes.

untagged NEC220, suggesting that the C-terminal His$_8$-tag has no deleterious effect on the membrane budding activity [19,31]. Previously, we showed that the NEC constructs that lack the MPRs do not bud membranes even when tethered to Ni-containing liposomes by His$_8$-tag because such a tag cannot restore the ability to order lipids [31].

By itself, the SUP$_{UL31}$ mutation did not change the budding efficiency of NEC220-His$_8$ (**Fig 4C**), similar to the untagged NEC220 (**Fig 1E**). As previously reported by our group, SE6$_{UL31}$ mutation reduced *in-vitro* budding to ~10% of the WT NEC220-His$_8$ despite the ability to interact with membranes due to the His$_8$-tag [31] (**Fig 4C**). Therefore, we performed a cryoEM analysis to examine NEC220-SE6$_{UL31}$-His$_8$ membrane interactions. NEC220-SE6$_{UL31}$-His$_8$ was incubated with large unilamellar vesicles (LUVs) of similar composition to the GUVs used for the budding assay and imaged with cryoEM (**Fig 4D**). NEC220-SE6$_{UL31}$-His$_8$ formed membrane-bound spikes on the outside of the LUVs (**Fig 4D**), but the internal protein coats indicative of budding [19,21,32] were rarely observed. This is reminiscent of the behavior of the oligomerization-deficient NEC-DN$_{UL34}$ mutant previously reported by our group [19]. We conclude that the SE6$_{UL31}$ mutations perturb NEC oligomerization, likely as the consequence

of weakened MPR/membrane interactions [31]. Surprisingly, the $SUP_{UL31}$ mutation restored budding of the $SE6_{UL31}$ mutant to ~50% of the WT NEC220-His$_8$ (**Fig 4C**). Therefore, the $SUP_{UL31}$ mutation can rescue budding *in vitro*, even if partially, to a mutant that indirectly disrupts oligomerization by weakening MPR/membrane interactions.

## WT NEC, NEC-SUP$_{UL31}$, and NEC-DN$_{UL34}$/SUP$_{UL31}$ form similar hexagonal lattices on vesicles budded *in vitro*

To identify the mechanism by which the $SUP_{UL31}$ mutation can restore budding to a broad range of mutants, we examined its effect on the NEC lattice. As the $SUP_{UL31}$ mutation rescues budding defects caused by mutations that disrupt hexagonal lattice interfaces, it would be expected to reinforce the weakened hexagonal lattice. Yet, residue $R229_{UL31}$ is far away from the hexameric interface and while near the interhexameric interface, it does not make any contacts there (**Fig 1C**). Therefore, we initially hypothesized that the $SUP_{UL31}$ mutation promoted the formation of an NEC lattice with an alternative geometry.

To examine the effect of the $SUP_{UL31}$ mutation on the geometry of the membrane-bound NEC coats, we performed cryoEM/T analyses on WT NEC220 and the mutants NEC220-SUP$_{UL31}$ and NEC220-DN$_{UL34}$/SUP$_{UL31}$. Each protein complex was incubated with LUVs similar in composition to those used in our prior cryoET reconstructions of the WT NEC220 coats [19,21,32]. Using sub-tomographic averaging, we obtained the 3D reconstructions of the WT NEC220 (5.9 Å), NEC220-SUP$_{UL31}$ (13.1 Å), and NEC220-DN$_{UL34}$/SUP$_{UL31}$ (5.4 Å) (**Fig 5**). The lower final resolution of the NEC220-SUP$_{UL31}$ reconstruction was due to the aggregation of budded vesicles formed by NEC220-SUP$_{UL31}$, which reduced the number of NEC220-SUP$_{UL31}$ particles available for data processing. The higher resolution of the WT NEC220 and NEC220-DN$_{UL34}$/SUP$_{UL31}$ averaged cryoET density map allowed us to dock the crystal structure of the WT NEC185Δ50 heterodimer (**Fig 5D and 5F**), confirming that the $SUP_{UL31}$ mutation does not perturb NEC conformation in a major way, even when bound to membranes. We also observed additional helical density at the C terminus of UL34, which corresponded to helix α4 that was unresolved in the WT NEC185Δ50 crystal structure [24] yet present in the crystal structures of NEC homologs from PRV [24,35], HCMV [25,36], and EBV [21]. The PRV UL34 α4 helix fit well into the HSV-1 UL34 cryoET averages (**Fig 5E and 5G**).

All three constructs formed very similar hexagonal lattices on budded membranes (**Fig 5A–5C** and **S1 Table**), showing that the $SUP_{UL31}$ mutation did not promote the formation of an NEC lattice with an alternative geometry. Based on the cryoET data, we hypothesized that the $SUP_{UL31}$ mutation reinforced the hexagonal lattice.

## WT NEC, NEC-SUP$_{UL31}$, and NEC-DN$_{UL34}$/SUP$_{UL31}$ form similar hexagonal lattices in the crystals

To obtain a high-resolution view of the hexagonal lattice formed by the NEC-SUP$_{UL31}$ mutant, we crystallized the NEC185Δ50-SUP$_{UL31}$ and the NEC185Δ50-DN$_{UL34}$/SUP$_{UL31}$ constructs, the mutant equivalents of the previously crystallized WT NEC185Δ50 construct (UL31: 51–306 and UL34: 15–185) [24]. Both mutants took the space group C2$_1$ with six crystallographically independent NEC heterodimers in the asymmetric unit, SUP$_{AB}$, SUP$_{CD}$, SUP$_{EF}$, SUP$_{GH}$, SUP$_{IJ}$, and SUP$_{KL}$ (UL34 chains: A, C, E, G, I, and K; UL31 chains: B, D, F, H, J, and L) (**S2 Table**). Phases were determined using molecular replacement with the WT NEC185Δ50 structure as a search model. The NEC185Δ50-SUP$_{UL31}$ structure was refined to 3.9-Å resolution (**S2 Table**). The atomic coordinates and structure factors of the NEC185Δ50-SUP$_{UL31}$ structure were deposited to the RCSB Protein Data Bank under the accession number 8G6D. However, the NEC185Δ50-DN$_{UL34}$/SUP$_{UL31}$ crystals diffracted x-rays only to ~6 Å resolution,

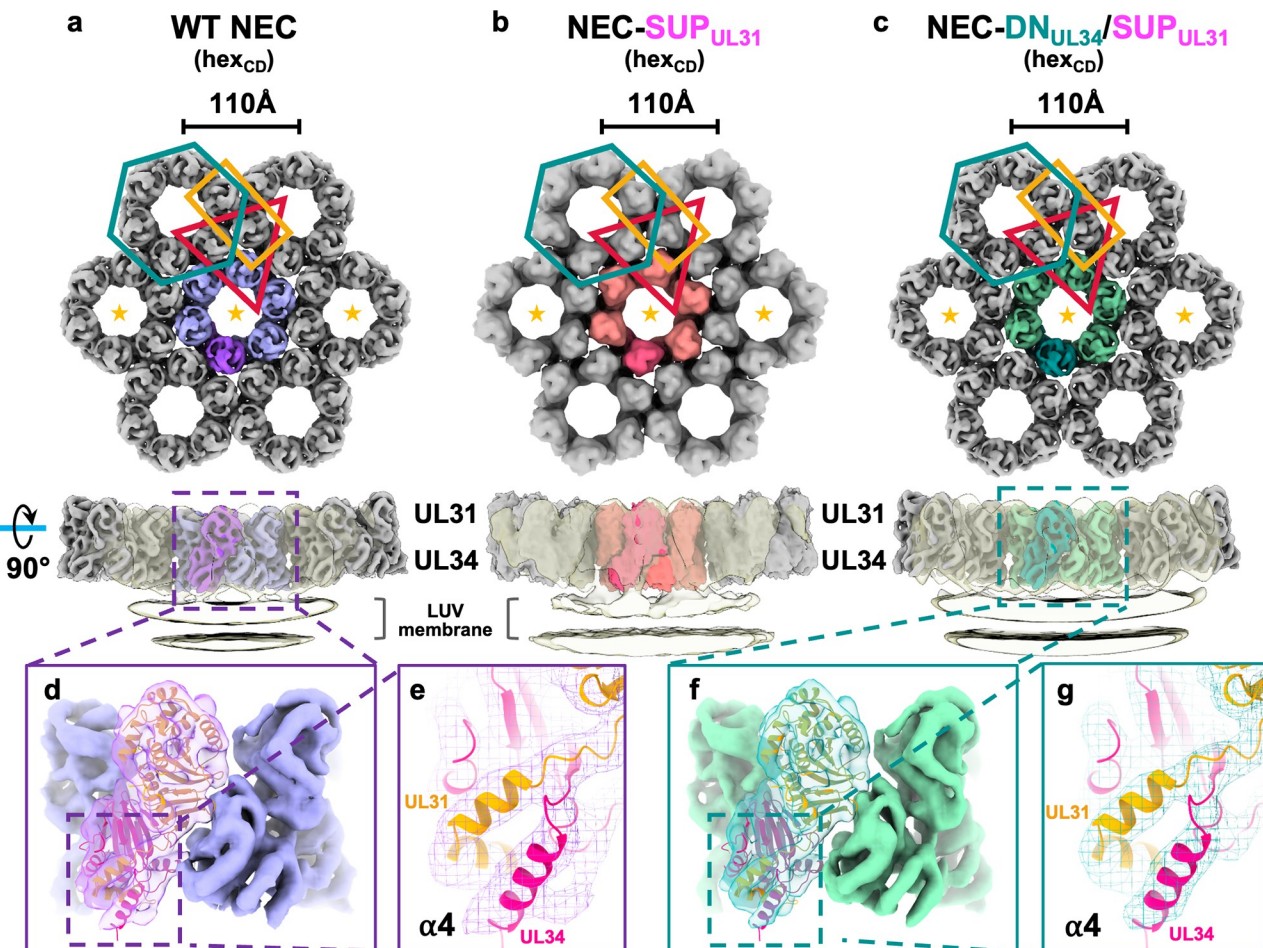

**Fig 5. WT NEC, NEC-SUP$_{UL31}$, and NEC-DN$_{UL34}$/SUP$_{UL31}$ form similar membrane-bound hexagonal coats.** CryoET reconstruction of **a)** the WT NEC coat at 5.9 Å, **b)** the NEC-SUP$_{UL31}$ coat at 13.1 Å, and **c)** the NEC-DN$_{UL34}$/SUP$_{UL31}$ coat at 5.4 Å. Only the three hexameric units marked with orange stars are shown in lower 90˚-rotated panels, where the low-pass filtered transparent densities show the connection between NEC lattices and LUV membrane. The HSV-1 NEC crystal structure (PDB ID: 4ZXS) docks similarly into both the WT NEC **(d)** and NEC-DN$_{UL34}$/SUP$_{UL31}$ **(f)** cryoET densities. **e, g)** Docking of the α4 helix from the PRV NEC crystal structure (PDB ID: 4Z3U) accounts for the additional density observed in both cryoET reconstructions which was originally unresolved in the HSV-1 NEC crystal structure [24]. All coats have the hex$_{CD}$ arrangement.

which precluded atomic refinement (**S2 Table**). Therefore, the NEC185Δ50-SUP$_{UL31}$ structure was used for the in-depth analysis of the interfaces and conformational changes whereas the structural analysis of the NEC185Δ50-DN$_{UL34}$/SUP$_{UL31}$ structure was limited to the analysis of crystal packing.

The previously crystallized HSV-1 WT NEC185Δ50 construct took the space group P6 with two crystallographically independent NEC heterodimers in the asymmetric unit, NEC$_{AB}$ and NEC$_{CD}$ (UL34 chains: A and C; UL31 chains: B and D) [24]. While the two heterodimers formed very similar, perfectly symmetrical hexamers, they arranged into two distinct hexagonal lattices, hex$_{AB}$ and hex$_{CD}$ (**Fig 6A and 6B and S3 Table**) [24]. In both lattices, hexamer interactions result in trimers formed by UL31/UL31 interactions (**Fig 6A and 6B, red**). However, the hex$_{AB}$ lattice also has two types of dimers formed by either UL31/UL31 or UL31/UL34 interactions (**Fig 6A, coral and gold,** respectively) whereas the hex$_{CD}$ lattice has only one dimer type formed by UL31/UL34 interactions (**Fig 6B, gold**).

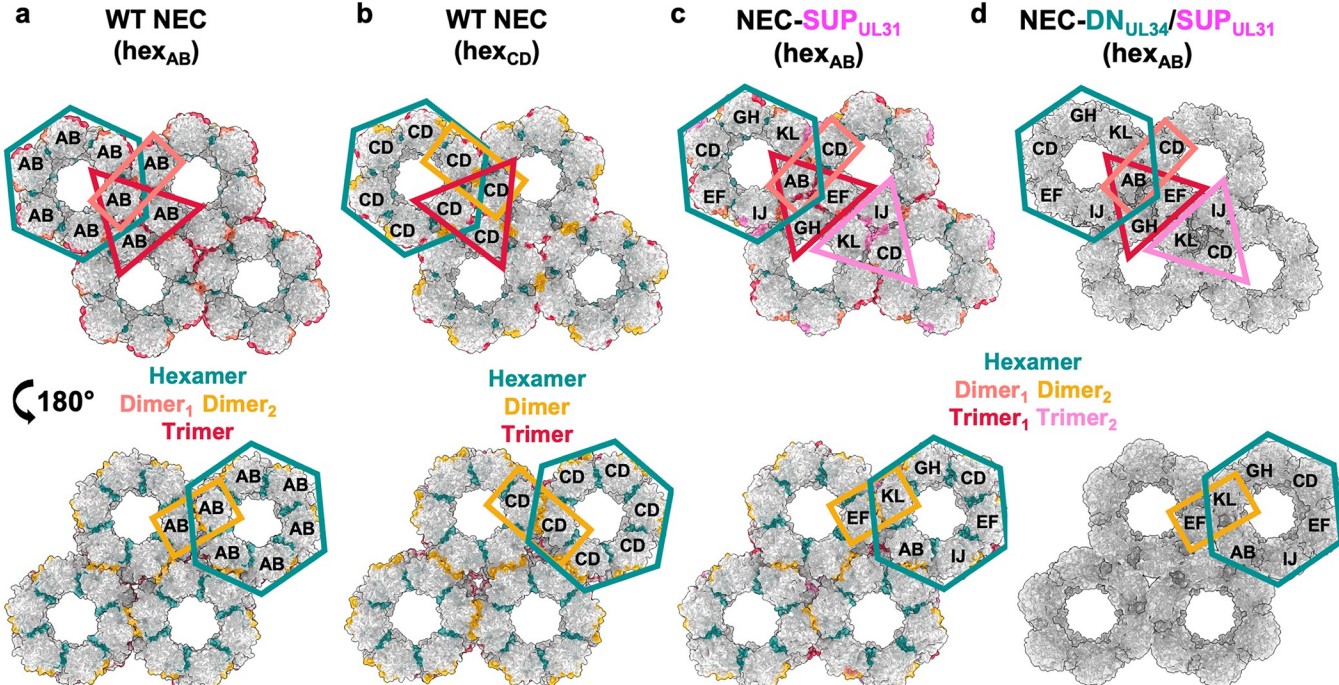

**Fig 6. NEC-SUP$_{UL31}$ and NEC-DN$_{UL34}$/SUP$_{UL31}$ form hex$_{AB}$ lattices in the crystals. a)** The HSV-1 WT NEC hex$_{AB}$ (PDB ID: 4ZXS)$_,$ **b)** the WT NEC hex$_{CD}$ (PDB ID: 4ZXS), **c)** the NEC-SUP$_{UL31}$ and **d)** the NEC-DN$_{UL34}$/SUP$_{UL31}$ crystal lattices [24]. Hexameric (teal) and interhexameric (dimer$_1$: coral, dimer$_2$: gold, trimer$_1$: red, and trimer$_2$: light pink) interfaces are colored accordingly. Two distinct trimers formed in the NEC-SUP$_{UL31}$ lattice (red and light pink). Due to resolution, an interface analysis was not performed on the NEC-DN$_{UL34}$/SUP$_{UL31}$ crystal lattice. The corresponding heterodimers within the lattice are labeled.

Just as the WT NEC185Δ50, the NEC185Δ50-SUP$_{UL31}$ formed hexamers in the crystals. Only in this case, the hexamers were not perfectly symmetrical, being formed by six independent, non-crystallographic heterodimers in the asymmetric unit (**Fig 6C**). Nonetheless, the NEC185Δ50-SUP$_{UL31}$ hexamers looked very similar to the WT NEC185Δ50 hexamers, with similar hexameric interfaces (**S3 and S4 Tables**). The crystal lattice formed by the NEC185Δ50-SUP$_{UL31}$ hexamers (**Fig 6C**) resembled the WT NEC hex$_{AB}$ lattice (**Fig 6A and S5–S7 Tables**). The NEC185Δ50-DN$_{34}$/SUP$_{31}$ crystal lattice (**Fig 6D**) also resembled the WT NEC hex$_{AB}$ lattice (**Fig 6A**). The SUP$_{UL31}$ mutation thus had no major effect on the hexamer structure or the hexagonal lattice.

Interestingly, while the WT NEC formed both hex$_{AB}$ and hex$_{CD}$ lattices in the crystals and the mutants formed only the hex$_{AB}$ lattice (**Fig 6**), in the membrane-bound coats, the WT NEC220 and both mutants formed the hex$_{CD}$ lattice (**Fig 5**). The reasons for these differences are yet unclear. Regardless, just as the WT NEC, the mutants could form both types of the hexagonal lattice.

## The SUP$_{UL31}$ mutation eliminates a salt bridge and changes loop conformations

To identify any local conformational changes in the mutant NEC lattices, we compared NEC185Δ50-SUP$_{UL31}$ heterodimers to each other and to the WT NEC185Δ50 heterodimers (**Fig 7**). The six crystallographically independent heterodimers in the NEC185Δ50-SUP$_{UL31}$ structure were structurally similar to each other (**S4 Fig and S8 and S9 Tables**) and to the two WT NEC heterodimers [24] (**Figs 7 and S4 and S10 Table**) and could be superimposed with

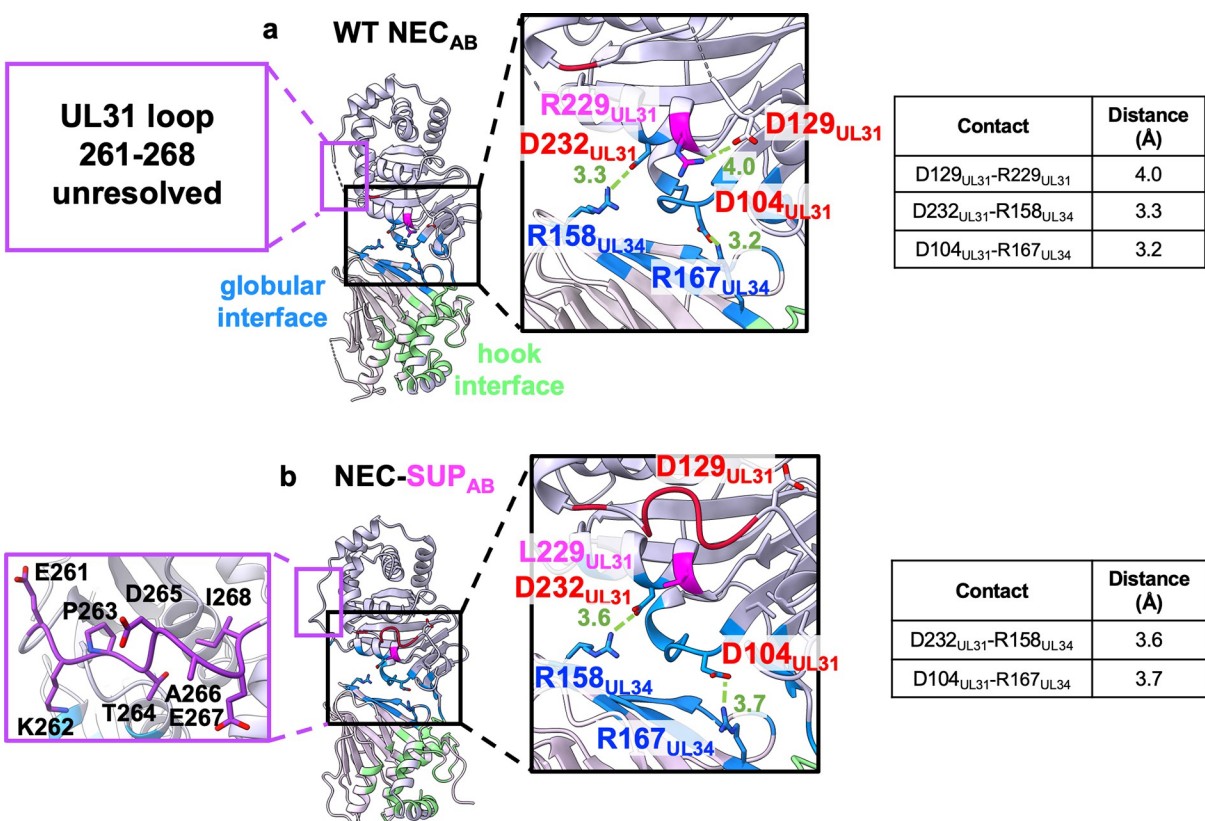

**Fig 7. The SUP$_{UL31}$ mutation eliminates a salt bridge near the heterodimeric interface and changes loop conformations.** The crystal structures of **(a)** WT NEC$_{AB}$ and **(b)** NEC-SUP$_{AB}$ heterodimers. The heterodimeric interface residues are colored blue (globular interface) or green (hook interface). Residue 229$_{UL31}$ is colored magenta. Insets show close-up views of the globular heterodimeric interface. Residues forming salt bridges at the interface are shown as sticks and colored in blue (Rs) or red (Es and Ds). Residue 229$_{UL31}$ is shown as sticks and colored in magenta. Salt bridges are shown as dashed green lines, with distances in Angstrom. Distances are also listed in the corresponding tables. The resolved portions of the dynamic loops 129$_{UL31}$-134$_{UL31}$ and 261$_{UL31}$-268$_{UL31}$ are shown in red and purple, respectively. The HSV-1 NEC crystal structure (PDB: 4ZXS) was used to generate the figure in (**a**).

RMSDs ranging from 0.67 to 1.02 Å (**S11 Table**). Thus, the SUP$_{UL31}$ mutation did not alter the overall NEC structure.

Nonetheless, we detected several local conformational changes. First, all six copies of UL34 in the NEC-SUP$_{UL31}$ mutant contained an additional density at the C terminus (**S4A Fig**), corresponding to a portion of the helix α4 that was unresolved in the WT NEC185Δ50 crystal structure [24] but present in the crystal structures of NEC homologs [21,24,25,35–37].

Second, two loops that were partially unresolved in the WT structures due to disorder were resolved in the NEC-SUP$_{UL31}$ mutant: residues 129$_{UL31}$-134$_{UL31}$ (resolved in SUP$_{AB}$, SUP$_{CD}$, SUP$_{EF}$, and SUP$_{IJ}$; partially resolved in SUP$_{GH}$) and 261$_{UL31}$-268$_{UL31}$ (resolved in all six heterodimers) (**Figs 7 and S4B and S6 Table**. The 129$_{UL31}$-134$_{UL31}$ loop was located directly above residue 229$_{UL31}$ that, in turn, was located directly above the heterodimeric globular interface (**Figs 7 and S4B**). In the WT NEC$_{AB}$ structure, the R229$_{UL31}$ side chain made a salt bridge with the nearby D129$_{UL31}$ located at one end of the mostly disordered 129$_{UL31}$-134$_{UL31}$ loop (**Fig 7A**). The R229L$_{UL31}$ mutation eliminated this salt bridge, and in all SUP$_{UL31}$ heterodimers (except for SUP$_{KL}$ where it was unresolved), the D129$_{UL31}$ side chain pointed away from L229$_{UL31}$ (**Figs 7B and S4B**). The lack of the R229$_{UL31}$-D129$_{UL31}$ salt bridge in the NEC-SUP$_{UL31}$ structure correlated with an ordered 129$_{UL31}$-134$_{UL31}$ loop (**Fig 7B**). We hypothesize

that by eliminating the salt bridge, the $SUP_{UL31}$ mutation released the loop, which allowed it to adopt an ordered conformation. Due to the lack of obvious contacts between residue $229_{UL31}$ and the $261_{UL31}$-$268_{UL31}$ loop, we postulate that the $SUP_{UL31}$ mutation promoted ordering of the $261_{UL31}$-$268_{UL31}$ loop through a long-range, allosteric mechanism.

## The $SUP_{UL31}$ mutation generates new contacts at the interhexameric interface

To determine whether the $SUP_{UL31}$ mutation changed any contacts at the lattice interfaces, we analyzed the buried surface areas and sidechain contacts (hydrogen bonds and salt bridges) in the WT NEC and NEC-$SUP_{UL31}$ crystal structures using PDBePISA interface analysis [38]. We found that 5 out of 6 hexameric interfaces in the NEC-$SUP_{UL31}$ crystal lattice had buried surface areas reduced by ~15% compared to the WT NEC $hex_{AB}$ and $hex_{CD}$ lattices (**S12 Table**). These hexameric interfaces also had fewer hydrogen bonds and salt bridges (**S13 Table**) whereas the remaining hexameric interface, A/K, had more contacts than the WT (**S13 Table**). However, the interhexameric interfaces in the NEC-$SUP_{UL31}$ crystal lattice had increased buried surface areas (**S14 Table**) and new contacts (**S15 Table**). Importantly, some of the new contacts at the interhexameric interface were mediated by the $129_{UL31}$-$134_{UL31}$ and $261_{UL31}$-$268_{UL31}$ loops that were ordered in the NEC-$SUP_{UL31}$ but not the WT NEC structures (**S6 Table**).

A prominent example of new contacts at the interhexameric interface were several new salt bridges (**Fig 8**). The trimeric interface in the WT NEC $hex_{AB}$ lattice had one salt bridge,

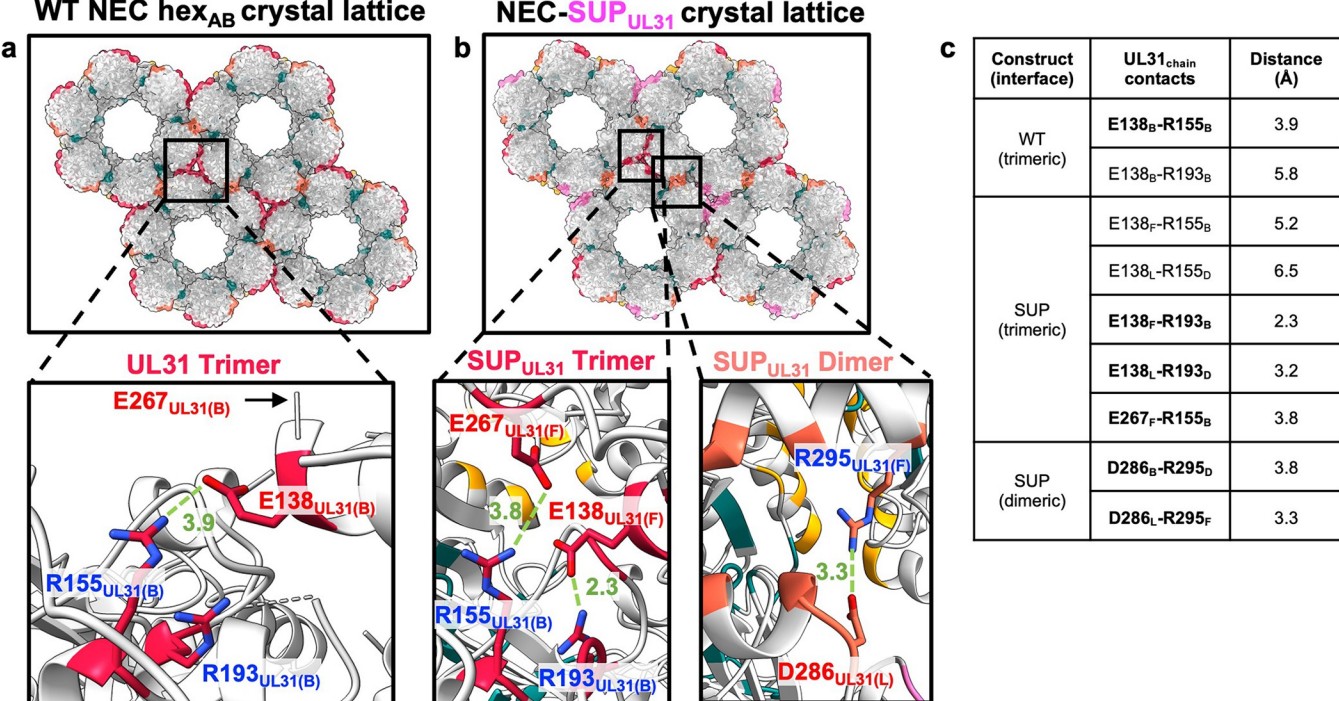

**Fig 8. The interhexameric interfaces in the NEC-$SUP_{UL31}$ mutant have new salt bridges.** Close-up views of the interhexameric interfaces in the (**a**) WT NEC crystal lattice (UL31 trimer; crimson) and (**b**) NEC-$SUP_{UL31}$ crystal lattice [UL31 trimers (crimson and pink) and dimers (orange and yellow)]. The HSV-1 NEC crystal structure (PDB: 4ZXS) was used to generate the figure in panel **a**. Salt bridges are shown as green dashed lines, with distances in Angstrom. Residues forming salt bridges are shown as sticks and colored in blue (Rs) or red (Es or Ds). **c**) Distances of contacts at the highlighted interfaces. Contacts in bold are salt bridges.

| Construct (interface) | UL31$_{chain}$ contacts | Distance (Å) |
|---|---|---|
| WT (trimeric) | **E138$_B$-R155$_B$** | 3.9 |
| | E138$_B$-R193$_B$ | 5.8 |
| SUP (trimeric) | E138$_F$-R155$_B$ | 5.2 |
| | E138$_L$-R155$_D$ | 6.5 |
| | **E138$_F$-R193$_B$** | 2.3 |
| | **E138$_L$-R193$_D$** | 3.2 |
| | **E267$_F$-R155$_B$** | 3.8 |
| SUP (dimeric) | **D286$_B$-R295$_D$** | 3.8 |
| | **D286$_L$-R295$_F$** | 3.3 |

E138$_{UL31}$-R155$_{UL31}$ (**Fig 8A**). But in the NEC-SUP$_{UL31}$ lattice, E138$_{UL31}$ and R155$_{UL31}$ formed new salt bridges with other residues. E138$_{UL31}$ formed a salt bridge with R193$_{UL31}$ at both distinct trimeric interfaces, B/F/H and D/J/L (**Fig 8B**). R155$_{UL31}$ formed a salt bridge with E267$_{UL31}$, but only in the B/F/H trimer (**Fig 8B**) (in the D/J/L trimer, R155$_{UL31}$ and E267$_{UL31}$ were ~7 Å apart). In the WT NEC$_{AB}$ structure, the unresolved E267$_{UL31}$ was too far away from the interhexameric interface to participate in any contacts (**Fig 8A**). Moreover, there was another new salt bridge, D286$_{UL31}$-R295$_{UL31}$, at the dimeric interfaces, B/D and F/L (**Fig 8B**), which was absent from the WT NEC lattice (**Fig 8A**). Therefore, in the NEC-SUP$_{UL31}$ mutant, there were new contacts at the interhexameric interface that could, in principle, stabilize the NEC lattice in the presence of lattice-destabilizing mutations. We hypothesize that new interhexameric contacts also form in the hex$_{CD}$ lattice, but in the absence of higher resolution data, residues that participate in these contacts remain unknown.

## Discussion

Herpesviruses translocate their capsids from the nucleus to the cytoplasm by an unusual mechanism that requires the formation of membrane-bound coats by the virally encoded heterodimeric complex, the NEC [19,22,23]. The coats are composed of a hexagonal NEC lattice, and mutations that disrupt the lattice interfaces reduce budding *in vitro* [19,24,31] and viral replication [26–28,34]. The hexagonal lattice thus plays a central role in the NEC membrane-budding function. Here, we demonstrated that a suppressor mutation within the UL31 protein, SUP$_{UL31}$, acts as a universal suppressor of membrane budding defects in NEC because it restored efficient membrane budding *in vitro* and viral replication to a broad range of budding-deficient NEC mutants. We hypothesize that the SUP$_{UL31}$ mutation exerts its powerful suppressor effect by reinforcing the hexagonal lattice destabilized by mutations. Indeed, we found that the hexagonal lattice formed by the NEC-SUP$_{UL31}$ maintains a similar geometry yet has new interface contacts. The increased interface would be expected to reinforce the hexagonal NEC lattice, helping it to counteract the lattice-destabilizing effects of mutations. We hypothesize that its dynamic, flexible nature allows the NEC lattice to adapt to perturbations that it likely encounters during nuclear egress.

### The SUP$_{UL31}$ mutation promotes the formation of new interhexameric contacts

Since the SUP$_{UL31}$ mutation rescued budding defects caused by disruptions of the hexagonal lattice, we initially hypothesized that it may do so promoting the formation of a different, potentially, non-hexagonal lattice with distinct interfaces. Instead, we found that NEC-SUP$_{UL31}$ and NEC-DN$_{UL34}$/SUP$_{UL31}$ mutants formed hexagonal lattices both in the crystals and in membrane-bound coats that were very similar to those formed by the WT NEC. However, the interhexameric interfaces in the NEC-SUP$_{UL31}$ crystal lattice are larger due to several new interactions, particularly, salt bridges. How could the SUP$_{UL31}$ mutation promote new contacts at the interhexameric interface? Residue L229$_{UL31}$ itself does not participate in any interface contacts. Instead, we hypothesize that it promotes new contacts through a combination of direct and allosteric mechanisms. By eliminating the R229$_{UL31}$-D129$_{UL31}$ salt bridge (**Fig 7**), the SUP$_{UL31}$ mutation would release the 129$_{UL31}$-134$_{UL31}$ loop, allowing the latter to adopt an ordered conformation and form new contacts at the interhexameric interface (**Fig 8B**). In the absence of obvious contacts between residue 229$_{UL31}$ and the 261$_{UL31}$-268$_{UL31}$ loop, ordering of the latter probably occurred through a long-range, allosteric mechanism. Larger interhexameric lattice interfaces would be expected to reinforce the lattice. By stabilizing the lattice disrupted by mutations, the SUP$_{UL31}$ mutation could restore efficient budding.

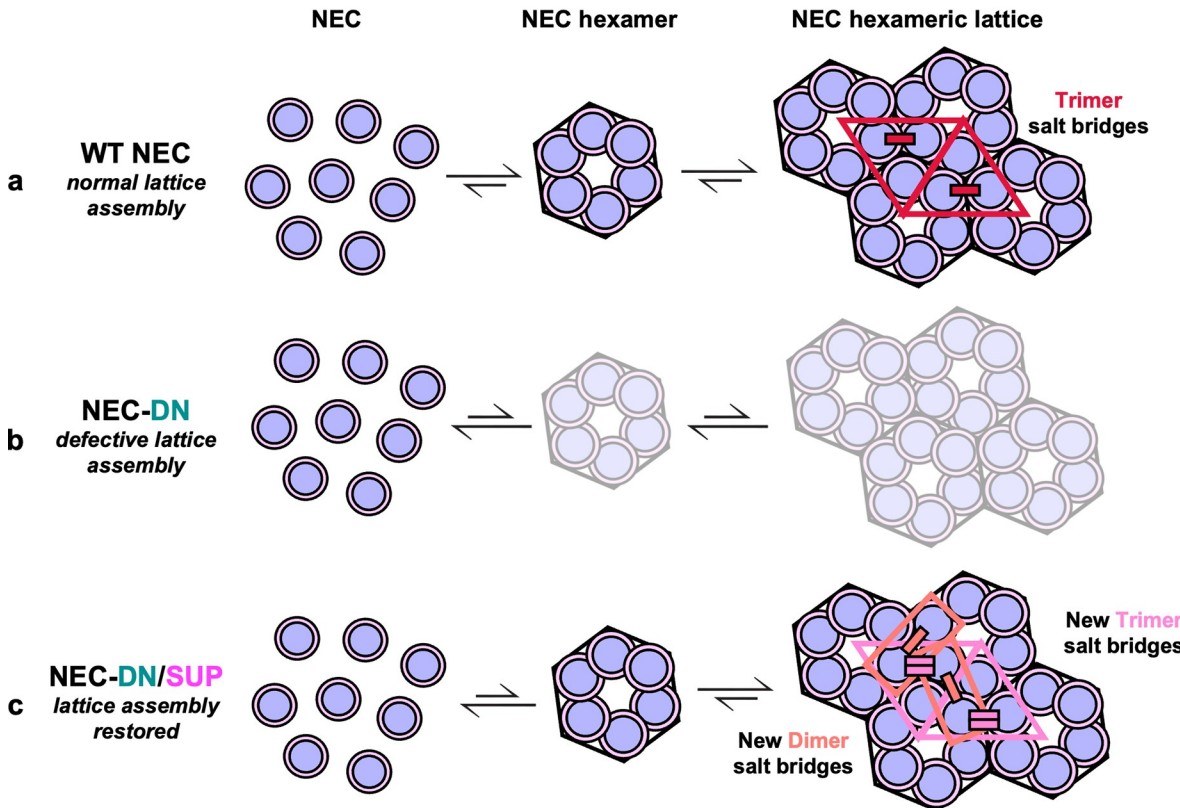

Fig 9. A model of SUP$_{UL31}$ budding restoration in the context of a lattice destabilizing mutation. a) WT NEC heterodimers are arranged into hexamers that assemble into a stable hexagonal lattice by forming contacts at the hexameric and interhexameric interfaces. The salt bridges located at the trimeric interhexameric interfaces (crimson) favor lattice association, rather than disassociation. (b) In the presence of a lattice destabilizing mutation such as NEC-DN$_{UL34}$, hexamer formation and lattice assembly are perturbed. (c) The SUP$_{UL31}$ mutation restores hexagonal lattice formation by promoting the formation of new contacts at both the dimeric and trimeric interhexameric interfaces (pink and salmon), resulting in a stable and functional NEC lattice despite the presence of a destabilizing mutation.

It is easy to envision how the SUP$_{UL31}$ mutation might suppress budding defects caused by mutations that destabilize interhexameric interactions by compensating for the loss of those interactions locally with new interhexameric interactions. However, the SUP$_{UL31}$ mutation also suppresses budding defects of mutants that destabilize the hexamers themselves. Therefore, we propose a more general mechanism for this type of suppression. We hypothesize that the NEC hexamers weakened by interface mutations could be stabilized not only by the interactions between adjacent NEC heterodimers within the hexamer itself but also by their incorporation into a larger lattice where interhexameric contacts would limit the dissociation of NEC heterodimers from the hexamer (Fig 9). By strengthening these latter contacts, the SUP$_{UL31}$ mutation could, in principle, compensate for different kinds of lattice defects.

## The SUP$_{UL31}$ mutation could make the NEC more conformationally dynamic

The WT NEC185Δ50 crystallized in the P6 space group, in which the hexamers are perfect, being related by crystallographic symmetry. However, both the NEC185Δ50-SUP$_{UL31}$ and

NEC185$\Delta$50-DN$_{UL34}$/SUP$_{UL31}$ mutants crystallized in the C2$_1$ space group, in which the hexamers are non-crystallographic and, thus, imperfect. This could imply that the NEC heterodimer became more conformationally dynamic in the presence of the SUP$_{UL31}$ mutation. Indeed, the heterodimeric UL31/UL34 interface between the globular cores of UL31 and UL34 buried a ~6–20% larger surface area in all six NEC-SUP$_{UL31}$ heterodimers compared to the WT NEC heterodimers (**S16 Table**), suggesting flexibility at this interface. Experimental or computational analysis of the NEC dynamics await further studies outside the scope of the current work.

Residue 229$_{UL31}$ is located between two intermolecular salt bridges at the heterodimeric interface formed by the UL31 and UL34 globular cores, R158$_{UL34}$-D232$_{UL31}$ and R167$_{UL34}$-D104$_{UL31}$ (**Figs 7 and S4**). Previous molecular dynamics (MD) study proposed that these salt bridges contribute to the overall stability of the HSV-1 NEC heterodimer [39]. The HCMV NEC, which has only one salt bridge, was found to be more dynamic than HSV-1 in MD simulations [39]. The EBV NEC, which also has only one salt bridge, is a conformationally dynamic heterodimer as revealed by its crystal structure [21]. The number of intermolecular salt bridges at the globular UL31/UL34 interface thus correlates with the stability of the NEC heterodimer.

We propose that the intramolecular salt bridge R229$_{UL31}$-D129$_{UL31}$ is another important contributor to the stability of the NEC heterodimer. Although the NEC-SUP$_{UL31}$ still has two intermolecular salt bridges at the heterodimeric interface, it lacks the intramolecular salt bridge (**Figs 7 and S4**), which increases its flexibility. This increased flexibility could explain why both NEC-SUP$_{UL31}$ and NEC-DN$_{UL34}$/SUP$_{UL31}$ required extra additives to form crystals relative to the WT NEC [24] and why both NEC-SUP$_{UL31}$ and NEC-DN$_{UL34}$/SUP$_{UL31}$ took the lower symmetry space group C2$_1$, rather than P6.

## The NEC forms two distinct hexagonal lattices with unclear biological significance

Thus far, two types of the hexagonal lattice have been observed, hex$_{AB}$ and hex$_{CD}$. HSV-1 NEC formed hex$_{AB}$ (WT-NEC, NEC-SUP$_{UL31}$, and NEC-DN$_{UL34}$/SUP$_{UL31}$) or hex$_{CD}$ (WT-NEC) lattice configurations in crystals and only hex$_{CD}$ lattice in membrane-bound coats. In the case of other crystallized NEC homologs, the HCMV NEC formed a hexagonal lattice resembling the hex$_{CD}$ lattice configuration in crystals [25] whereas the PRV NEC [24,35], the VZV NEC [37], and the EBV NEC [21] did not. Of note, the EBV NEC formed a yet undefined oligomeric assembly on membranes, possibly, due to its observed flexibility [21].

We do not yet know what dictates the choice of a particular lattice for a given NEC homolog or variant under given conditions (e.g., crystals vs. membrane) nor the relative biological significance of the two lattices. One potential reason for the apparent preference for the hex$_{CD}$ lattice on membranes by HSV-1 NEC *in vitro* could be that the cryoET NEC lattice averages, generated from NEC budded vesicles, represent the "end-product" of budding. In this scenario, the hex$_{AB}$ lattice could represent a lattice formed at an intermediate stage of budding, which would not be captured in our cryoET experiments. However, PRV NEC formed a hex$_{AB}$ lattice in budded vesicles formed in uninfected cells overexpressing the NEC [22] and in infected cells [40]. Therefore, it cannot be concluded that the hex$_{CD}$ lattice is preferred on membranes by the NEC or that it is more relevant biologically. We hypothesize that each NEC homolog can form both lattices, perhaps, under different experimental conditions, and that the ability to form two distinct lattices could be due to the inherent plasticity of the NEC. Regardless, how the NEC assembles into a hexagonal lattice of either type is still unknown.

## The $SUP_{UL31}$ mutation acts as a universal suppressor against mutations disrupting the NEC budding activity

The $SUP_{UL31}$ mutation was initially identified as an extragenic suppressor of a nuclear budding defect caused by a double mutation within UL34, $D35A_{UL34}/E37A_{UL34}$ ($DN_{UL34}$), in HSV-1 infected cells [27]. Subsequently, we showed that the $DN_{UL34}$ mutation blocks the formation of the hexagonal NEC lattice [19] by eliminating important polar contacts at the hexameric interface [24] and that the $SUP_{UL31}$ mutation restored effective membrane budding *in vitro* to the $DN_{UL34}$ and another hexameric interface mutant, $V92F_{UL34}$ [24]. Here, we found that the $SUP_{UL31}$ mutation could restore budding to many lattice interface mutants ($T123Q_{UL34}$, $F252Y_{UL31,}$ and $E153R_{UL31}$; **Fig 1**); the heterodimeric interface mutants ($K137A_{UL34}$ and $K137A/R139A_{UL34}$; **Fig 3**); and even to a membrane interface mutant ($SE6_{UL31}$; **Fig 4**). Thus, the $SUP_{UL31}$ mutation acts as a universal suppressor mutation that restores efficient budding *in vitro* and, in several cases, viral replication to a diverse range of budding-deficient NEC mutants.

Although mutations that cause budding defects target diverse interfaces, all are expected to destabilize the hexagonal NEC lattice, which is essential for the membrane budding process. The destabilizing effect of the lattice interface mutations is the most apparent. But mutations destabilizing the NEC heterodimer would also be expected to weaken the lattice by destabilizing its core building block. Finally, NEC/membrane interactions likely destabilize the lattice within the membrane-bound NEC coat indirectly. By reinforcing the lattice, the $SUP_{UL31}$ mutation could overcome these lattice destabilizing defects regardless of their nature.

In some cases, the $SUP_{UL31}$ mutation could fully restore the *in-vitro* budding activity but not viral replication, e.g., in the presence of the $T123Q_{UL34}$ and $E153R_{UL31}$ (**Fig 2**). We hypothesize that these mutations may perturb the NEC functions that do not involve membrane deformation, e.g., nuclear lamina dissolution, capsid docking at the INM, or capsid recruitment. Thus, the $SUP_{UL31}$ mutation specifically restores budding defects.

If the $SUP_{UL31}$ mutation forms a stronger hexagonal lattice, why isn't this mutation positively selected? We hypothesize that this mutation may impair other functions of the NEC. This idea is supported by the observation that in the *in-trans* complementation experiment, viral replication in the presence of the $SUP_{UL31}$ mutation does not reach the WT levels (**Fig 2**). While the $SUP_{UL31}$ mutation is not positively selected, it provides the virus with a strategy to maintain replication in the presence of external stressors such as inhibitors targeting the NEC. Further work to identify and characterize other herpesviral suppression mechanisms could aid in the advancement of novel antiherpesviral therapeutics.

## A dynamic, flexible lattice may be necessary for NEC function

During nuclear egress, the NEC generates sufficient curvature to wrap around an egressing capsid. The formation of the hexagonal NEC lattice is important for its function in capsid budding. Yet, a perfectly hexagonal lattice is incompatible with spherical curvature. To generate a spherical particle, a hexagonal lattice must contain lattice "defects", either regular defects, namely, insertions of a different geometry, or irregular defects. Both types of defects have been observed in other viral assembly systems. For example, icosahedral capsids, which are built of hexons, achieve curvature by incorporating pentons at the vertices. A similar strategy could be employed by HSV-1 through the formation of NEC pentamers, which we previously observed *in vitro* [32]. The studies herein add to the growing number of examples of varied NEC oligomeric states and lattice configurations, each of which may be used at varying points during viral replication, as observed for other viruses.

HIV-1 structural protein Gag, which forms the viral capsid and mediates viral budding, presents an interesting case. It assembles into a hexagonal lattice that has irregular defects in the immature HSV-1 capsid [41] but regular, pentameric insertions in the mature capsid [42]. In this way, Gag can "tune" its hexagonal lattice to perform a specific function. Moreover, Gag can form different types of hexagonal lattice. The capsid and matrix domains of Gag each adopted two distinct hexagonal lattices in immature and mature virions, each with a distinct functional role [43,44]. Thus, HSV-1 Gag exemplifies both the conformational dynamics and functional versatility of viral assembly lattices.

Likewise, the hexagonal NEC lattice is conformationally flexible. A recent study that visualized the NEC lattice in cells infected with a mutant pseudorabies virus (PRV), an alphaherpesvirus, by using the focused ion beam/scanning electron cryomicroscopy (cryoFIB/SEM), identified regions of disorder within the hexagonal NEC coats surrounding the capsid [40]. This observation suggests that the NEC lattice can incorporate irregular defects. In addition to irregular defects, non-hexameric NEC assemblies, such as pentamers [32] and heptamers [23], have been observed, albeit not in the context of a wild-type HSV-1 infection. Finally, the NEC can adopt different arrangements of the hexagonal lattice, either by the WT NEC (WT-hex$_{AB}$ and WT-hex$_{CD}$) or by the NEC-SUP$_{UL31}$ mutant presented in this study. We hypothesize that the ability to adopt different configurations allows the NEC to "tune" its lattice to assemble the coats and maintain their integrity in response to perturbations that occur during nuclear egress, such as changes in membrane curvature and interactions with binding partners.

## Materials and methods

### Cells and viruses

Vero and Hep-2 cells were maintained as previously described [11]. The properties of HSV-1 (F) and vRR1072(TK+) (a UL34-null virus derived by homologous recombination with HSV-1 (F) have also been previously described [11,45]. The UL34-null virus and the UL34-null/ SUP$_{UL31}$ recombinant viruses used for complementation assays were derived from the pYE-bac102 clone of the HSV-1 strain (F) genome in the bacterial strain GS1783 (a gift from G. Smith, Northwestern U.) [46–48] as previously described. All UL34-null viruses were propagated on Vero tUL34 CX cells that express HSV-1 pUL34 under the control of its native promoter regulatory sequences [49]. Vero tUL34 CX cells were propagated in DMEM high glucose supplemented with 5% fetal bovine serum and the antibiotic penicillin and streptomycin.

### Cloning

All primers used for cloning are listed in S17 Table. Cloning of UL31 (1–306), UL34 (1–220), UL34 (15–185), UL34-His$_8$ (1–220 with a C-terminal His$_8$-tag) and the corresponding UL31 and UL34 mutants [R229L$_{UL31}$ (SUP$_{UL31}$), D35A$_{UL34}$/E37A$_{UL34}$ (DN$_{UL34}$), E37A$_{UL34}$, T123Q$_{UL34}$, F252Y$_{UL31}$, and E153R$_{UL31}$, and S11E$_{UL31}$/S24E$_{UL31}$/S26E$_{UL31}$/S27E$_{UL31}$/ S40E$_{UL31}$/S43E$_{UL31}$ (SE6$_{UL31}$)] was previously described [19,24,31].

### Oligomeric interface mutants

Site-directed mutagenesis of pJB14 (UL31 1–306 R229L$_{UL31}$) was performed using splicing-by-overlap extension protocol followed by restriction digest into the pKH90 plasmid (containing an N-terminal His-SUMO-PreScission tag in-frame with a BamHI restriction site) to create either the F252Y$_{UL31}$/R229L$_{UL31}$ (pED20) or E153R$_{UL31}$/R229L$_{UL31}$ (pED21) double mutants.

### Heterodimeric interface mutants

Site-directed mutagenesis of pJB02 (UL34 1–220) was performed using splicing-by-overlap-extension protocol followed by restriction digest into the pJB02 plasmid (containing an N-terminal GST-PreScission tag in-frame with a SalI restriction site) to create either the $K137A_{UL34}$ (pED25), $R139A_{UL34}$ (pED26), or the $K137A_{UL34}$/$R139A_{UL34}$ (pED27) mutants.

### Membrane interface mutants

Site-directed mutagenesis of pJB60 ($UL31$-$SE6_{UL31}$ 1–306) was performed using an inverse PCR protocol followed by blunt-end ligation to create the $SE6_{UL31}$/$R229L_{UL31}$ mutant (pED45).

### Crystallization constructs

Digested PCR fragments containing $R229L_{UL31}\Delta50$–306 were amplified from pJB114 (UL31 1–306 $R229L_{UL31}$) and subcloned by restriction digest into a pET24b(+) plasmid harboring an N-terminal $His_6$-SUMO-PreScission tag in-frame with a BamHI restriction site plasmid to create the $R229L_{UL31}\Delta50$–306 plasmid (pXG20). Digested PCR fragments containing $D35A_{UL34}$/$E37A_{UL34}$ were amplified from pJB06 (UL34 1–246 $D35A_{UL34}$/$E37A_{UL34}$) and subcloned by restriction digest into a pGEX-6P1 vector containing an N-terminal GST-PreScission tag in-frame with a SalI restriction site to create the UL34 15–185 $D35A_{UL34}$/$E37A_{UL34}$ plasmid (pJB66).

### Cell complementation constructs

Plasmid pRR1072Rep, which was the parent vector for UL34 mutant plasmids used for cell culture complementation assays has been previously described [28]. Mutant derivatives of pRR1072Rep that carry the $D35A_{UL34}$/$E37A_{UL34}$ and $K137A_{UL34}$/$R139A_{UL34}$ double mutations have also been previously described and were referred to as CL04 and CL10 in that publication [28]. Derivatives of pRR1072Rep containing the single $D35A_{UL34}$, $E37A_{UL34}$, $T123Q_{UL34}$, $K137A_{UL34}$, and $R139A_{UL34}$ were constructed by Gibson assembly. Plasmids were assembled from two PCR products, each generated using pRR1072Rep as a template and using either a mutagenic forward primer paired with a reverse primer from the ampicillin resistance gene, or a mutagenic reverse primer paired with a forward primer from the ampicillin resistance genes. PCR products were digested with DpnI to remove template sequences and then assembled using the New England BioLabs 2X Gibson assembly master mix according to the manufacturer's instructions.

### Expression and purification of WT NEC, oligomeric interface, and heterodimeric interface mutants

Plasmids encoding HSV-1 UL31 1–306 (pKH90) and UL34 1–220 were co-transformed into *Escherichia coli* BL21(DE3) LoBSTr cells (Kerafast) to generate wild-type NEC220 [19,32]. All mutant constructs contained UL31 1–306 and UL34 1–220 amino acid boundaries. Plasmids encoding the appropriate mutations of either UL31 or UL34 were also co-transformed into *E. coli* BL21(DE3) LOBSTR cells (Kerafast) to generate the various NEC oligomeric and heterodimeric interface mutants (listed in **S18 Table**). The expression and purification of NEC220 and some oligomeric interface mutants (NEC-$DN_{UL34}$, NEC-$DN_{UL34}$/$SUP_{UL31}$, NEC-$SUP_{UL31}$, NEC-$E37A_{UL34}$, NEC-$T123Q_{UL34}$, NEC-$F252Y_{UL31}$, and NEC-$E153R_{UL31}$) were described previously [19,24]. The expression and purification of oligomeric interface mutants (NEC-$SUP_{UL31}$/$F252Y_{UL31}$, NEC-$SUP_{UL31}$/$E153R_{UL31}$, and NEC-$SUP_{UL31}$/$T123Q_{UL34}$) and

heterodimeric interface mutants (NEC-K137A$_{UL34}$, NEC-R139A$_{UL34}$, NEC-K137A$_{UL34}$/ R139A$_{UL34}$, NEC-K137A$_{UL34}$/SUP$_{UL31}$ and NEC-K137A$_{UL34}$/R139A$_{UL34}$/SUP$_{UL31}$) are described below. Cells expressing the corresponding NEC construct were expressed using auto-induction at 37˚C in TB supplemented with 100 µg/mL ampicillin, 100 µg/mL kanamycin, and 34 µg/mL chloramphenicol, 0.2% lactose, and 2 mM MgSO$_4$ for 4 h. The temperature was reduced to 25˚C for 16 h. Cells were harvested at 5,000 x g for 30 min.

All purification steps were performed at 4˚C, as previously described [19,32]. Cell pellets were resuspended in lysis buffer (50 mM Na HEPES pH 7.5, 500 mM NaCl, 1 mM TCEP, and 10% glycerol) supplemented with Complete protease inhibitor (Roche) and lysed using a microfluidizer (Microfluidics). The cell lysate was clarified by centrifugation at 18,000 x g for 40 min and passed over a Ni Sepharose 6 column (Cytiva) equilibrated with lysis buffer. The protein-bound column was washed with 20 mM and 40 mM imidazole lysis buffer and bound proteins were eluted with 250 mM imidazole lysis buffer. Eluted proteins were passed over a Glutathione Sepharose 4B column and washed with lysis buffer. The His$_6$-SUMO and GST tags were cleaved for 16 h by PreScission Protease, produced in-house from a GST-PreScission fusion expression plasmid (a gift of Peter Cherepanov, Francis Crick Institute). The protein was passed over 2 x 1 mL HiTrap Talon columns (Cytiva) to remove His$_6$-SUMO, followed by injection onto a size-exclusion column (Superdex 75 10/300; Cytiva) equilibrated into gel-filtration buffer (20 mM Na HEPES, pH 7.0, 100 mM NaCl, and 1 mM TCEP), as previously described. Fractions containing pure protein, as assessed by 12% SDS-PAGE and Coomassie staining were pooled and concentrated as described below.

For both NEC-SUP$_{UL31}$ and NEC-DN$_{UL34}$/SUP$_{UL31}$, the cleaved proteins were passed over a HiTrap SP XL (5 mL; Cytiva) ion-exchange column, to remove free His$_6$-SUMO. Bound proteins were eluted using a linear salt gradient (60 mL) made from no salt gel filtration buffer (20 mM Na HEPES, pH 7.0, and 1 mM TCEP) and salt gel filtration buffer (20 mM Na HEPES, pH 7.0, 1 M NaCl, and 1 mM TCEP). Proteins typically eluted ~ 360 mM NaCl, at which point the gradient was held constant until the UV signal returned to baseline. Fractions containing pure protein, as assessed by 12% SDS-PAGE and Coomassie staining, were pooled and diluted using no salt gel filtration buffer to reach a 100 mM NaCl concentration, which is required for downstream liposome budding experiments described below. For all purifications described herein, the protein was concentrated to ~ 1 mg/mL and stored at -80˚C to prevent degradation observed at 4˚C. Protein concentrations were determined by the absorbance at 280 nm. A typical yield was ~0.5 mg/L.

## Expression and purification of membrane interface constructs

Plasmids containing UL31 1–306 (pKH90) and UL34 1-220-His$_8$ (pJB57) were co-transformed into *E. coli* BL21(DE3) LoBSTr cells (Kerafast) to generate NEC220-His$_8$. All the following constructs contained UL31 1–306 and UL34 1–220 amino acid boundaries. A list of plasmids co-transformed to create the NEC-SE6$_{UL31}$-His$_8$, NEC-SUP$_{UL31}$-His$_8$, and NEC-SE6$_{UL31}$/ SUP$_{UL31}$-His$_8$ constructs are listed in **S18 Table**. Cells expressing the corresponding NEC mutant were grown using auto-induction at 37˚C in TB supplemented with 100 µg/mL ampicillin, 100 µg/mL kanamycin, and 34 µg/mL chloramphenicol, 0.2% lactose, and 2 mM MgSO$_4$ for 4 h. The temperature was reduced to 25˚C for 16 h. Cells were harvested at 5,000 x g for 30 min.

Cells were resuspended in lysis buffer supplemented with Complete protease inhibitor (Roche) and lysed using a microfluidizer (Microfluidizer). The cell lysate was clarified by centrifugation at 18,000 x g for 40 min and passed over a Ni Sepharose 6 column (Cytiva). The column was washed with 20 mM and 40 mM imidazole lysis buffer and bound proteins were

eluted with 250 mM imidazole lysis buffer. Eluted proteins were passed over a Glutathione Sepharose 4B column and washed with lysis buffer. The $His_6$-SUMO and GST tags were cleaved for 16 h by PreScission Protease, produced in-house from a GST-PreScission fusion expression plasmid. The protein was loaded on a size-exclusion column (Superdex 75 10/300; Cytiva) equilibrated with a gel-filtration buffer. Fractions containing pure protein, as assessed by 12% SDS-PAGE and Coomassie staining were pooled and concentrated as described above.

## Expression and purification of crystallization constructs

$SUP_{UL31}\Delta50$–306 (pXG20) and UL34 15–185 (pJB04) or $SUP_{UL31}\Delta50$–306 (pXG20) and UL34 15–185 $D35A_{UL34}/E37A_{UL34}$ (pJB66) plasmids were co-transformed into *E. coli* BL21(DE3) LoBSTr cells (Kerafast) to produce the NEC-$SUP_{UL31}$ and NEC-$DN_{UL34}/SUP_{UL31}$ crystallization constructs, respectively. Cells expressing the corresponding NEC mutant were grown using auto-induction at 37˚C in TB supplemented with 100 μg/mL ampicillin, 100 μg/mL kanamycin, and 34 μg/mL chloramphenicol, 0.2% lactose, and 2 mM $MgSO_4$ for 4 h. The temperature was reduced to 25˚C for 16 h. Cells were harvested at 5,000 x g for 30 min.

Cells were resuspended in lysis buffer supplemented with Complete protease inhibitor (Roche) and lysed using a microfluidizer (Microfluidizer). The cell lysate was clarified by centrifugation at 18,000 x g for 40 min and passed over a Ni Sepharose 6 column (Cytiva). The column was washed with 20 mM and 40 mM imidazole lysis buffer and bound proteins were eluted with 250 mM imidazole lysis buffer. Eluted proteins were passed over a Glutathione Sepharose 4B column and washed with lysis buffer. The $His_6$-SUMO and GST tags were cleaved for 16 h by PreScission Protease, produced in-house from a GST-PreScission fusion expression plasmid. The protein was passed over 2 x 1 mL HiTrap Talon columns (Cytiva) to remove $His_6$-SUMO, followed by injection onto a size-exclusion column (Superdex 75 10/300; Cytiva) equilibrated into gel-filtration buffer, as previously described. Fractions containing pure protein, as assessed by 12% SDS-PAGE and Coomassie staining were pooled and concentrated as described above.

## *In-vitro* budding assays

Giant unilamellar vesicles (GUVs) were prepared as previously described [19]. For budding quantification, 10 μL of POPC:POPA:POPS = 3:1:1 (Avanti Polar Lipids) GUVs containing 0.2% ATTO-594 DOPE (ATTO-TEC GmbH) fluorescent dye were added to gel filtration buffer containing 0.2 mg/mL (final concentration) Cascade Blue Hydrazide (ThermoFisher Scientific) and either 1 μM WT NEC or NEC mutant (final concentration), for a total sample volume of 100 μL. Reactions incubated for 5 min at 20˚C before imaging in 96-well chambered coverglass (Brooks Life Science Systems). Samples were imaged using a Nikon A1R Confocal Microscope with a 60x oil immersion lens at the Imaging and Cell Analysis Core Facility at Tufts University School of Medicine. Budding events were quantified by manually counting ~300 vesicles total in 15 different frames of the sample. Before analysis, the background (GUVs in the absence of NEC) was subtracted from the raw values. All data values are reported in the **S1 Data**. Each sample was tested in at least three biological replicates, each containing three technical replicates. Reported values represent the average budding activity relative to NEC220 or NEC220-$His_8$ (100%). The standard error of the mean is reported for each measurement. Significance was calculated using an unpaired one-tailed *t*-test against NEC220. Statistical analyses and data presentation were performed using GraphPad Prism 9.1.0.

## Complementation assays

24-well cultures of Hep-2 cells at 70% confluence were transfected with 0.05 μg of pCMVβ, expressing the β-galactosidase gene, and 0.25 μg of wild-type or mutant UL34 plasmid using

Lipofectamine as described by the manufacturer (Gibco-BRL) and incubated at 37°C overnight. The cells were then infected with 10 PFU of the BAC-derived UL34-null virus or UL34-null/SUP$_{UL31}$ virus per cell and incubated at 37°C for 90 min. Monolayers were washed once with pH 3 sodium citrate buffer (50 mM sodium citrate, 4 mM potassium chloride, adjusted to pH 3 with hydrochloric acid) and then incubated at room temperature in fresh citrate buffer for one minute. Cells were washed with V medium (Dulbecco's modified Eagle's medium, penicillin-streptomycin, 1% heat-inactivated calf serum) two times. One milliliter of V medium was then added to each well, and after 16 h of incubation at 37°C, cell lysates were prepared by freezing and thawing followed by sonication for 20 seconds at power level 2 with a Fisher sonic dismembrator. The amount of infectivity in each lysate was determined by plaque assay titration on UL34-complementing cells. Part of each cell lysate was assayed for β-galactosidase expression as previously described [28]. Transfection efficiencies in all samples were within 20% of each other. Each sample was tested in at least three biological replicates, each containing one technical replicate. The raw titers and log PFU values for each biological replicate are reported in the **S1 Data**.

## Evaluation of mutant protein expression in mammalian cells

12-well cultures of Hep-2 cells at 70% confluence were transfected with 0.5 μg of wild-type or mutant UL34 or UL31-FLAG plasmid using Lipofectamine as described by the manufacturer (Gibco-BRL) and incubated at 37°C overnight. The cells were then infected with 10 PFU of the corresponding null virus and incubated at 37°C for 90 min. The inoculum was then removed and replaced with 1.5 ml V medium, and cultures were incubated for a further 16 hours. Infected cells were harvested by removing the medium, washing the monolayers once with phosphate-buffered saline (PBS), scraping the cells into 1 ml PBS, and pelleting the cells at 3,000 rpm in the microcentrifuge for 3 minutes. The supernatant liquid was removed, and cell lysates were prepared by resuspending the cell pellets in 25 μl water, adding 25 μl of 2X SDS-polyacrylamide gel sample buffer, and then incubating in a boiling water bath for 10 minutes. Proteins were separated by SDS-PAGE, blotted to nitrocellulose, and then probed with chicken antibody to UL34 (diluted 1:250) [11], mouse antibody to HSV-1 VP5 (diluted 1:500 (Biodesign International) mouse antibody to FLAG epitope (diluted 1:1000) (monoclonal M2, SIGMA/Aldrich) or rabbit antibody to calnexin (diluted 1:1000) (Cell Signaling Technology).

## Crystallization and data collection

Crystals of NEC185-Δ50-SUP$_{UL31}$ were grown by vapor diffusion at 25°C in hanging drops with 1 μL of protein (3 mg/mL), 1 μL of reservoir solution (10% PEG3350, 8 mM Li$_2$SO$_4$, 6 mM ATP, and 0.1 M MES, pH 6) and 1 μL of Silver Bullets (Hampton Research) reagent [G3 (0.25% 2,2'-Thiodiglycolic acid, 0.2% Azelaic acid, 0.2% Mellitic acid, 0.2% trans-aconitic acid, 0.02 M HEPES sodium pH 6.8)]. Hexagonal SUP$_{UL31}$ crystals appeared after 2 days, only in the presence of Silver Bullets, and were completely grown after one week. Crystals were flash-frozen into liquid nitrogen in a solution identical to the reservoir solution containing 30% glycerol as the cryoprotectant.

Crystals of NEC185-Δ50-DN$_{UL34}$/SUP$_{UL31}$ were grown by vapor diffusion at 25°C in hanging drops with 1 μL of protein (3.5 mg/mL), 1 μL of reservoir solution (10% PEG3350, 14 mM Li2SO4, 14 mM ATP, 10 mM phenol, and 0.1 M MES, pH 6) and 1 μL of Silver Bullets (Hampton Research) reagent [G3 (0.25% 2,2'-Thiodiglycolic acid, 0.2% Azelaic acid, 0.2% Mellitic acid, 0.2% trans-aconitic acid, 0.02 M HEPES sodium pH 6.8)]. Hexagonal DN$_{UL34}$/SUP$_{UL31}$ crystals appeared after one week, only in the presence of Silver Bullets. Crystals were flash-frozen into liquid nitrogen in a solution identical to the reservoir solution containing 30%

glycerol as the cryoprotectant. In comparison to the $SUP_{UL31}$ crystals, $DN_{UL34}/SUP_{UL31}$ crystals required an additional additive, phenol, and took longer to appear (one week vs. two days).

### CryoEM grid preparation and image collection

A volume of 10 μL of a 1:1 mixture of 400 nm and 800 nm large unilamellar vesicles (LUVs) made of 60% POPC/10% POPS/10% POPA/10% POPE/5% cholesterol/5% Ni-DGS [prepared as previously described [19]] were mixed at room temperature with $NEC220$-$SE6_{UL31}$-$His_8$ to a final protein concentration of 1 mg/mL. After 15 min incubation, 3 μL of the sample was applied to glow-discharged (45 s) Quantifoil (2/2; Electron Microscopy Sciences) grids. Grids were blotted on both sides for 6 s with 0 blotting force and vitrified immediately by plunge freezing into liquid nitrogen-cooled liquid ethane (Vitrobot), before storage in liquid nitrogen. Grids were loaded into a Tecnai F20 transmission electron microscope (FEI) with an FEG and Compustage, equipped with a Gatan Oneview CMOS camera, using a cryo holder (Gatan) (Brandeis University Electron Microscope Facility). The microscope was operated in low-dose mode at 200 keV with SerialEM. Images were recorded at 19,000-fold (pixel size: 5.6 nm) magnification at a defocus of -4 μm. Images are displayed using ImageJ [50].

### CryoET grid preparation

A volume of 10 μL of a 1:1 mixture of 400 nm and 800 nm large unilamellar vesicles (LUVs) made of POPC:POPS:POPA = 3:1:1 [prepared as previously described [19]] was mixed on ice with either WT-NEC220, $NEC$-$SUP_{UL31}$, or $NEC$-$DN_{UL34}/SUP_{UL31}$, each to a final protein concentration of 1 mg/mL. After 30 min incubation, the sample was mixed with 5 nm fiducial gold beads, and cryoET grids were prepared by applying 3 μL of sample to glow-discharged (30 s) Lacey carbon grids (Electron Microscopy Sciences). Grids were blotted on both sides for 6 s with 0 blotting force and vitrified immediately by plunge freezing into liquid nitrogen-cooled liquid ethane with an FEI Mark IV Vitrobot cryosample plunger. Vitrified cryoET grids were stored in a liquid nitrogen dewar until use.

### CryoET data collection and tomogram reconstruction

Tilt series were collected a Titan Krios electron microscope at the California NanoSystems Institute (CNSI). Data collection parameters are listed in **S11 Table**. Tilt series were collected using SerialEM [51] in a Titan Krios instrument equipped with a Gatan imaging filter (GIF) and a post-GIF K3 direct electron detector in electron-counting mode. Frames in each movie of the raw tilt series were aligned, drift-corrected, and averaged with Motioncor2 [51]. The tilt series micrographs were aligned and reconstructed into 3D tomograms using the IMOD software package [52], then missing-wedge corrected by IsoNet [53] for particle picking.

### Sub-tomographic averaging

The variation in curvature of the NEC hexagonal coat made it difficult to identify the hexagonal repeat units required for particle picking. To overcome this, particle picking was performed using Python scripts derived from the Particle Estimation for Electron Tomography (PEET) software [54]. Firstly, an initial model was generated as previously described [55] by manually picking ~100 particles and performing sub-tomogram averaging using PEET. This allowed for the hexagonal geometric parameter, including the repeating distance and orients, of the NEC lattice to be accurately measured. Secondly, for each tomogram, a small set of particles were manually picked as "seed" particles sparsely covering all areas containing NEC. The

"seed" particle set was then expanded by adding unknown particles near each of the known particles based on the hexagonal geometry obtained above. PEET alignment was performed on the expanded particles to match local conformational changes. Finally, the particle set expansion and PEET alignment were performed iteratively to obtain a complete particle set. Particles with less than three neighbors were excluded from the final particle set to remove outliers. Coordinates and orientations of the final particle set were formatted and imported into Relion [56] for further processing. One round of 3D refinement under bin4 pixel size and several rounds of 3D refinement and classification under bin2 pixel size, along with duplicate removal, resulted in the final masked resolutions: WT NEC (5.9 Å), NEC-DN$_{UL34}$/SUP$_{UL31}$ (5.4 Å), and NEC-SUP$_{UL31}$ (13.1 Å). The resolutions reported above for the averaged structures are based on the 'gold standard' refinement procedures and the 0.143 Fourier shell correlation (FSC) criterion (**S5 Fig**).

### 3D visualization

UCSF ChimeraX [57] was used to visualize the resulting sub-tomogram averages in their three dimensions, segmentation of density maps, and surface rendering for the different components of NEC.

### Supporting information

**S1 Fig. Expression levels of WT UL31, WT UL34 and corresponding mutant proteins in Hep-2 cells used for *trans* complementation assays presented in this study.** Hep-2 cells were transfected with either wild-type or mutant UL34 **(a)** or UL31-FLAG **(b)** plasmids following by infection with the corresponding null virus. After incubation, cells were harvested, lysed, and analyzed via western blotting with the corresponding antibodies. VP5 and calnexin proteins were used as positive controls.
(PDF)

**S2 Fig. SDS-PAGE analysis of fractions from size-exclusion chromatography (Superdex S75). a)** NEC-R139A$_{UL34}$, **b)** NEC-K137A$_{UL34}$, and **c)** NEC-K137A$_{UL34}$/R139A$_{UL34}$. Fractions containing equimolar amounts of UL31 and UL34 (blue), unequal amounts of UL31 and UL34 (green), and free UL34 (magenta) are boxed. Only fractions containing equimolar amounts of UL31 and UL34 (blue) were pooled for use in downstream studies. UL31 is ~34 kDa and UL34 is ~25 kDa.
(PDF)

**S3 Fig. SDS-PAGE analysis of fractions from size-exclusion chromatography (S75). a)** NEC-K137A$_{UL34}$/SUP$_{UL31}$ and **b)** NEC-K137A$_{UL34}$/R139A$_{UL34}$/SUP$_{UL31}$. Fractions containing equimolar amounts of UL31 and UL34 (blue), unequal amounts of UL31 and UL34 (green), and free UL34 (magenta) are boxed. Only fractions containing equimolar amounts of UL31 and UL34 (blue) were pooled for use in downstream studies. UL31 is ~34 kDa and UL34 is ~25 kDa.
(PDF)

**S4 Fig. The six mutant NEC-SUP$_{UL31}$ heterodimers are similar to each other and to the WT NEC heterodimers. a)** Secondary structure overlay of WT NEC$_{AB}$ and WT NEC$_{CD}$ to the six NEC-SUP$_{UL31}$ heterodimers. Colored circles indicate variable regions in UL31: aa 194–198 (green), 129–134 (peach), and 261–268 (dark purple). The light purple circle indicates additional UL34 C-terminal residues (aa 175–178) resolved in the NEC-SUP$_{UL31}$ heterodimers that were unresolved in the WT NEC structures. **b)** The crystal structures of the WT-NEC$_{CD}$, NEC-SUP$_{CD}$, NEC-SUP$_{EF}$, NEC-SUP$_{GH}$, NEC-SUP$_{IJ}$, and NEC-SUP$_{KL}$ heterodimers.

WT-NEC$_{AB}$ and NEC-SUP$_{AB}$ are shown in **Fig 7**. The position of either R229 in the WT NEC or L229 in the NEC-SUP is shown in magenta. Residues at the UL31/UL34 globular interface are shown in blue. Insets show the resolved portions of the dynamic loops 129$_{UL31}$-134$_{UL31}$ and 261$_{UL31}$-268$_{UL31}$ in peach and dark purple, respectively. The HSV-1 NEC crystal structure (PDB: 4ZXS) was used to generate the figure in **a)** and **b)**.
(PDF)

**S5 Fig. Directional Fourier shell correlation (FCS) curves for the subtomogram averages of NEC lattices. a)** WT NEC, **b)** NEC-SUP$_{UL31}$, and **c)** NEC-DN$_{UL34}$/SUP$_{UL31}$.
(PDF)

**S1 Table. Conditions used for cryoET data collection and processing of WT and mutant NEC coats in this study.**
(PDF)

**S2 Table. Data collection and refinement statistics for NEC185Δ50-SUP$_{UL31}$ and NEC185Δ50-DN$_{UL34}$/SUP$_{UL31}$ crystal structures.**
(PDF)

**S3 Table. Residues involved in hexameric interactions in the WT NEC$_{A/B}$, WT NEC$_{C/D}$, and NEC-SUP$_{UL31}$ lattices.** Interfaces between UL31 and UL34 (boxes shaded in teal) and between UL34 and UL34 (boxes shaded in dark green) were analyzed using PDBePISA analysis [38]. Residues unresolved in the structures are indicated as NR.
(PDF)

**S4 Table. Conservation of residues at the hexameric interfaces within the NEC-SUP$_{UL31}$ lattice relative to the WT NEC$_{A/B}$ and WT NEC$_{C/D}$ lattices.** Interfaces were analyzed using PDBePISA analysis [38].
(PDF)

**S5 Table. Conservation of residues at the interhexameric interfaces within the NEC-SUP$_{UL31}$ lattice relative to the WT NEC$_{A/B}$ and WT NEC$_{C/D}$ lattices.** Interface residues were determined using PDBePISA analysis [38].
(PDF)

**S6 Table. Residues involved in interhexameric (trimeric) interactions in the WT NEC$_{A/B}$, WT NEC$_{C/D}$, and the NEC-SUP$_{UL31}$ lattices.** Interface residues (boxes shaded in light orange) were analyzed using PDBePISA analysis [38]. Residues unresolved in the structures are indicated as NR.
(PDF)

**S7 Table. Residues involved in interhexameric (dimeric) interactions in the WT NEC$_{A/B}$, WT NEC$_{C/D}$, and the NEC-SUP$_{UL31}$ lattices.** Interface residues are shaded in light orange (Dimer 1: UL31/UL31 and UL34/UL34) and dark orange (Dimer 2: UL31/UL31). Interfaces were analyzed using PDBePISA analysis [38].
(PDF)

**S8 Table. Resolved residues for each chain of UL34 (top) and UL31 (bottom) from the NEC185Δ50-SUP$_{UL31}$ crystal structure.** Resolved residues, along with % resolved, are listed for each of the chains.
(PDF)

**S9 Table. Structural alignments of the NEC-SUPUL31 heterodimers in the asymmetric unit.** The UL31/UL34 heterodimers and individual UL34 and UL31 chains were aligned.

RMSD values (Å) were calculated using "SSM Superpose" in WinCoot [59].
(PDF)

**S10 Table. Residues involved in heterodimeric interactions in the WT NEC$_{A/B}$, WT NEC$_{C/D}$, and the NEC-SUP$_{UL31}$ heterodimers.** Heterodimeric UL31/UL34 interfaces, the hook interface (boxes shaded light green) and the globular interface (boxes shaded in blue) were analyzed using PDBePISA analysis [1]. Residues unresolved in the structures are indicated as NR.
(PDF)

**S11 Table. Structural alignments of the NEC-SUP$_{UL31}$ to the WT NEC$_A/_B$ and WT NEC$_C/_D$ heterodimers.** NEC- SUP$_{UL31}$ heterodimers and individual UL34 and UL31 chains were aligned to the two WT NEC heterodimers and the corresponding WT individual chains. RMSD (Å) values are reported and were calculated using "SSM Superpose" in WinCoot [59].
(PDF)

**S12 Table. Buried surface areas at the hexameric interfaces.** PDBePISA analysis [38] was used to calculate the buried surface areas at the hexameric interfaces (either between UL31/UL34 or UL34/UL34) within the WT NEC$_{AB}$, WT NEC$_{CD}$, and NEC-SUP$_{UL31}$ lattices. The total buried surface area at the hexameric interface was calculated by adding the UL31/UL34 and UL34/UL34 surface areas. For the WT NEC, the RSCB PDB 4ZXS structure was used.
(PDF)

**S13 Table. Comparison of contacts made at the WT NEC and NEC-SUP$_{UL31}$ lattice hexameric interfaces.** Atomic contacts (hydrogen bonds or salt bridges) between heterodimers at the hexameric interfaces (shaded in blue) were analyzed using PDBePISA [38].
(PDF)

**S14 Table. Buried surface areas at the interhexameric interfaces.** PDBePISA analysis [38] was used to calculate the buried surface areas at the interhexameric interfaces (either between UL31 trimers or UL31 dimers) within the WT NEC$_{AB}$, WT NEC$_{CD}$, and NEC-SUP$_{UL31}$ mutant lattices. The total buried surface area at the trimeric interface was calculated by adding the UL31/UL31 surface areas within a trimer. For the WT NEC, the RCSB PDB 4ZXS structure was used.
(PDF)

**S15 Table. Comparison of contacts made at the interhexameric interfaces in the WT NEC and NEC-SUP$_{UL31}$ lattices.** Atomic contacts (hydrogen bonds or salt bridges) between heterodimers at the interhexameric interfaces (shaded in blue) were analyzed using PDBePISA [38].
(PDF)

**S16 Table. Buried interface surface areas for the globular, hook, and the total heterodimeric interface between UL31 and UL34.** PDBePISA analysis [38] was used to calculate the buried surface areas between the globular, hook, and total heterodimer interfaces within the WT NEC$_{AB}$, WT NEC$_{CD}$, and the six NEC-SUP$_{UL31}$ heterodimers. The globular interface area was determined by deleting the UL31 hook region (residues 54–88) prior to the PDBePISA analysis. The hook interface was determined by subtracting the globular area from the total heterodimeric interface area. The total heterodimeric interface area was calculated from the entire crystal structure. For WT NEC, the RSCB PDB 4ZXS structure was used.
(PDF)

**S17 Table. List of primers used for cloning procedures described in Materials and Methods.** All primers are listed in the 5'-3' direction. Restriction sites are underlined, and mutations

are bolded.
(PDF)

**S18 Table. List of plasmids used to create the NEC constructs used in this study not previously described.**
(PDF)

**S1 Data. Source file for the data used to generate Figs 1–4.**
(XLSX)

## Acknowledgments

We thank Janna Bigalke (Tufts University) for cloning the $SUP_{UL31}$, $SE6_{UL31}$, and UL34 15–185 constructs; Xuanzong Guo (Tufts University) for cloning the UL31Δ50–306 SUP plasmid and for purifying the $NEC-R139A_{UL34}/SUP_{UL31}$ and $NEC-D35A_{UL34}/SUP_{UL31}$ constructs. We also thank the staff at the NE-CAT (Advanced Photon Source) for help with collecting x-ray diffraction data; Peter Cherepanov (Francis Crick Institute) for the gift of the GST-PreScission Protease expression plasmid; and Thomas Schwartz (Massachusetts Institute of Technology) for the gift of LoBSTr cells. We thank Rob Jackson (Tufts University School of Medicine) and Martin Hunter (University of Massachusetts College of Engineering) for help with fluorescence microscopy experiments. Confocal microscopy was performed at the Imaging and Cell Analysis Core Facility within the Center for Neuroscience Research at Tufts University School of Medicine, which is supported by NIH grant P30 NS047243 (Rob Jackson). CryoEM samples were prepared and imaged at the Brandeis Electron Microscopy Facility. This work is based upon research conducted at the Northeastern Collaborative Access Team beamlines, which are funded by the National Institute of General Medical Sciences from the National Institutes of Health (P30 GM124165). The Eiger 16M detector on the 24-ID-E beamline is funded by an NIH-ORIP HEI grant (S10OD021527). This research used resources of the Advanced Photon Source; a U.S. Department of Energy (DOE) Office of Science User Facility operated for the DOE Office of Science by Argonne National Laboratory under Contract No. DE-AC02-06CH11357. All software was installed and maintained by SBGrid [58].

## Author Contributions

**Conceptualization:** Elizabeth B. Draganova, Ekaterina E. Heldwein.

**Data curation:** Elizabeth B. Draganova.

**Formal analysis:** Elizabeth B. Draganova, Hui Wang, Melanie Wu, Shiqing Liao, Gonzalo L. Gonzalez-Del Pino, Richard J. Roller, Ekaterina E. Heldwein.

**Funding acquisition:** Elizabeth B. Draganova, Z. Hong Zhou, Richard J. Roller, Ekaterina E. Heldwein.

**Investigation:** Elizabeth B. Draganova, Hui Wang, Melanie Wu, Shiqing Liao, Amber Vu, Gonzalo L. Gonzalez-Del Pino, Richard J. Roller.

**Methodology:** Elizabeth B. Draganova, Hui Wang, Shiqing Liao, Z. Hong Zhou, Richard J. Roller, Ekaterina E. Heldwein.

**Project administration:** Z. Hong Zhou, Richard J. Roller, Ekaterina E. Heldwein.

**Resources:** Z. Hong Zhou.

**Software:** Hui Wang.

**Supervision:** Z. Hong Zhou, Richard J. Roller, Ekaterina E. Heldwein.

**Validation:** Elizabeth B. Draganova, Hui Wang, Shiqing Liao, Richard J. Roller.

**Visualization:** Elizabeth B. Draganova, Hui Wang, Richard J. Roller.

**Writing – original draft:** Elizabeth B. Draganova, Ekaterina E. Heldwein.

**Writing – review & editing:** Elizabeth B. Draganova, Hui Wang, Melanie Wu, Z. Hong Zhou, Richard J. Roller, Ekaterina E. Heldwein.

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
