## [Decision Letter · Decision Letter 0]

16 Oct 2023

Dear Dr. Heldwein,

Thank you very much for submitting your manuscript "The universal suppressor mutation in the HSV-1 nuclear egress complex restores membrane budding defects by stabilizing the oligomeric lattice." for consideration at PLOS Pathogens. As with all papers reviewed by the journal, your manuscript was reviewed by members of the editorial board and by several independent reviewers. The reviewers appreciated the attention to an important topic. Based on the reviews, we are likely to accept this manuscript for publication, providing that you modify the manuscript according to the review recommendations.

Three experts in the field have reviewed your manuscript and were in strong agreement that this work is interesting and significant advance in the field. Please note that each reviewer also had requested changes to the text which are considered minor modifications, but important for the final consideration of the manuscript. Please address these minor modifications with detailed response letter in your revised manuscript.

Sincerely,

Paul M Lieberman

Academic Editor

PLOS Pathogens

Alison McBride

Section Editor

PLOS Pathogens

Kasturi Haldar

Editor-in-Chief

PLOS Pathogens

orcid.org/0000-0001-5065-158X

Michael Malim

Editor-in-Chief

PLOS Pathogens

orcid.org/0000-0002-7699-2064

Three experts in the field have reviewed your manuscript and were in strong agreement that this work is interesting and significant advance in the field. Please note that each reviewer also had requested changes to the text which are considered minor modifications, but important for the final consideration of the manuscript. Please address these minor modifications with detailed response letter in your revised manuscript.

Reviewer Comments (if any, and for reference):

Reviewer's Responses to Questions

**Part I - Summary**

Reviewer #1: This paper by Draganova et al. explores mechanisms by which suppressor mutations rescue the function of the HSV-1 NEC complex at the atomic level. The work is interesting as it sheds new light on structural considerations underlying NEC lattice formation and function. The work is thorough and generally well presented. That the suppressor mutation acts to stabilize the lattice by creation of new inter-haxameric binding fits well with its capability to suppress a variety of otherwise destabilizing mutations.

Reviewer #2: Draganova et al. present studies arising from a collaboration among the Heldwein, Roller, and Zhou laboratories that studies a mutation in the HSV-1 UL31 gene that was originally found to suppress a mutation in the UL34 gene. These two genes encode the subunits of the viral nuclear egess complex (NEC), whose structure was solved by the Heldwein lab. That structure did not readily explain the suppressor phenotype as the residues altered by the mutations were relatively far from each other. Here, the authors show that the suppressor mutation at least partially restores budding in an in vitro biochemical assay and, in some cases, viral replication in a transfected-infected cells to several different NEC mutants . These mutants either appear to have weaker NEC-NEC interactions for forming a hexagonal lattice or, in one case, are defective for interactions with membranes. Using crystallographic and cryo-EM tomography, the authors show that the suppressor effects are not due to major changes in a single NEC or formation of hexamers, but rather more subtle changes that include formation of new contacts between NEC hexamers within the lattice. Thus, the UL31 mutation has “universal suppressor” activity.

The various experiments in this study are well-designed and performed, the data are convincing, and the interpretations, for the most part, are sound. Moreover, the mechanism of suppression is cool and the authors make an interesting case for its relevance to potential anti-NEC therapies. The following comments are meant to improve the manuscript presentation and, in some cases, nudge the authors towards a more cautious take on their results.

Reviewer #3: The manuscript by Draganova et al provides an in-depth investigation of an interesting suppressor mutation in the HSV-1 nuclear egress complex (NEC). Some of the strengths of this study include the combination of robust biochemical budding assays, functional complementation assays and both crystallographic and cryo-EM tomography structural studies. Altogether, these approaches provide substantial data to gain a fuller picture of the impact of the SUP-UL31 mutation. These studies lead the authors to propose that the suppressor mutation (SUP-UL31) indirectly stabilizes the formation of hexagonal NEC assemblies at the membrane surface to overcome defects caused by deleterious mutations located at distant sites in the UL31/UL34 NEC. Overall, some of the strongest data for this comes from the in vitro budding assay where robust complementation of multiple defective mutants is observed with the SUP-UL31 mutant and where other potentially confounding factors in the viral complementation assays are absent. Generally, the manuscript is well written although some sections were a bit difficult to follow and I felt that some of the text discussing the structural data could be shortened.

**Part II – Major Issues: Key Experiments Required for Acceptance**

Reviewer #1: (No Response)

Reviewer #2: No key new experiments or modifications of existing experiments required.

Reviewer #3: The SUP-UL31 mutation restores/complements other viral budding mutants but this complementation in the virus titer assays does not reach wt levels - the SUP-UL31 mutant shows reduced titers relative to wt in its own right. This duality makes things a bit more complicated and raises the possibility that other factors are contributing. That said, the in vitro budding assay results are highly consistent with the interpretation that the authors settle on – as the complementation is much more direct and comparable to wt in this assay. I think the authors could discuss this more thoroughly in the final section of the manuscript and provide some explanation and/or evidence for alternative functions of UL31 that might impact virus production but not the in vitro budding assay.

Have the authors considered any approaches to more directly measuring the strength of the interhexameric interactions? Is there any evidence for a concentration-dependent assembly in solution that could be followed by light scattering? What about following the concentration dependence of 2D lattice formation on supported lipid bilayers?

In general, I felt that the section describing the virus complementation assays (lines 181-234) was difficult to follow and could be rewritten. If the mutant nomenclature could be simplified that might help but I think focusing the text on the direct comparisons that the authors would like to make in the data would also be helpful. The key comparisons that are drawn are comparing the relative effects of transfected mutants (e.g. UL34 mutants) relative to wt across two different virus constructs, which can be confusing to follow (see the next comment for more specific detail).

Line 226: “the SUP-UL31 mutation suppressed the complementation defects of both the D35A-UL34/E37A-UL34 227 and E37A-UL34 mutants similarly to the WT UL34 (Fig. 2c). “ I found this phrasing confusing since it relies on a comparison with the relative effects of the mutants in the wt and SUP-UL31 virus backgrounds, across different panels in Fig. 2. I think it may help to first focus on more explicitly explaining how each mutant performs relative to wt in each of the panels and then draw the relative comparisons across the wt and SUP-UL31 viruses. For example, for this line I would say that what is shown is that the D35AUL34/E37AUL34 227 and E37AUL34 mutants behave similarly to wt UL34 in the SUP-UL31 background, which is not the case in the WT-UL31 background. But – and I think this is important to state clearly - in the SUP-UL31 background, using wt UL34 does not yield the same virus titers obtained with wt UL31 virus. The reduced titers that are obtained with the SUP-UL31 virus also reduce the ability to discriminate effects of the mutants. For example in Fig. 2c, wt UL34 only increases titers by ~10-fold, providing a significantly reduced window to observe the effects of other UL34 mutations. I think the authors need to point out these issues clearly in the accompanying text in this secton.

Line 327: “Previously, we showed that such artificial tethering does not override the requirement for the MPR/membrane interactions and does not restore budding to MPR mutants with budding defects.” Do the authors have an explanation for the lack of an effect of the his-tag since this could be expected to enhance membrane recruitment and assembly? Why would it not complement the SE6 mutations but SUP-UL31 does?

The combination of X-ray and EM tomography is a strength of the analysis of the effect of the SUP-UL31 mutation on the formation of NEC hexamers. But highlights differences between the formation of the different HexAB vs HexCD lattices in truncated proteins in crystals vs full length on membranes. Any speculation as to why the Hex-CD form might be preferred on the membranes?

Lines 449-498: Section analyzing the inter-hexameric interfaces. This is a bit confusing since it seems more relevant to understand the Hex-CD interfaces given that these form preferentially on the membranes. How does this detailed analysis of the Hex-AB interfaces relate to what happens on membranes? Is there evidence that both hexamer assemblies form in vivo?

I think these sections and the analysis of the hex-AB form could be shortened given that the possibility that the more relevant form could be the Hex-CD assembly. Is there any way to model the role of residue 229 in the Hex-CD assembly, even at low resolution? I think it could be possible to combine last two sections of the paper into one (more concise) section. Some of the associated figures could be moved to supplemental data to contract Fig 7 and 8 into 1 figure that highlights the main points. I think it would be important include the caveat that conclusions on the role of the SUP-UL31 mutation for the Hex-CD form might be different.

Lines 559-593: “The SUP-UL31 mutation makes the NEC more conformationally dynamic.” I am not convinced that the mutant crystal structures are good evidence for increased flexibility in the NEC. The structural differences could easily be due to the lattice constraints in the two different crystal forms rather than the mutational difference. The fact that the NEC assemblies on the lipids also form a different hexameric form further suggest that one should be cautious in interpreting the crystal structures and their relationship to NEC dynamics.

**Part III – Minor Issues: Editorial and Data Presentation Modifications**

Reviewer #1: The font used in figure 2 below each panel should be changed to improve readability.

Line 203.-204. If UL31 F252Y is expressed to similar levels as other UL31’s, does it fail to complement? Suggesting that its ability to complement is attributable to overexpression is not well supported. They complement well even when expressed at physiological levels (i.e. when placed in recombinant virus).

Reviewer #2: 1. Title and elsewhere. Mutations are in genes, not in proteins. Although it can be a bit clumsy, the authors should reword.

2. Abstract. It would be helpful for the reader if the authors more completely summarized the experiments performed and their results rather than just mainly state the conclusions.

3. Introduction, first paragraph. Do we know that, for example, molluscan herpesviruses use the same kind of nuclear egress process as herpesviridae? If not, then perhaps just refer to herpesviridae here. Also, lines 63-71 of this paragraph lack any citations to the literature or even textbooks.

4. Fig. 1 legend, line 164. The number of asterisks here don’t conform with what panel 1E seems to show as the in the panel, the mutants showing what looks like the most significant effects have three asterisks, not two.

5. Line 194. Is this result really surprising? The mutant had only a ~3-fold effect in the biochemical assay and it appears to be overexpressed relative to WT in S1 Fig..

6. Fig. 2’s statistics require a correction for multiple comparisons. As the authors appropriately used corrections in other figures, perhaps this was just an oversight in drafting the legend, but if not, it may require modifying the figure and possibly some of the conclusions.

7. The complementation assay is very nice, but do the authors know whether they’re only looking at complementation rather than recombination to generate WT virus? For example, have they plaqued their virus on non-permissive cells, too?

8. Lines 268-270 and S2bc Fig. Why does free UL34 elute earlier from the size-exclusion column than the NEC when it would be expected to be smaller than the NEC. And, if it hadn’t already been stated, do these mutations alter residues near where UL31 and UL34 interact with each other?

9. Line 280, Fig. 3. Is there any hint of quantitative differences in heterodimer stability? If the assay can’t show such differences, the possibility that there could be such differences should be stated.

10. Line 368 and S4 Fig.. This is a key point, and might be emphasized more by putting it in a text figure and foreshadowing the subsequent discussion.

11. Lines 458-472 and elsewhere. The authors could be more cautious about interpreting differences in crystal structures and how they apply to what happens in solution and/or cells. There are many examples of crystal contacts being either irrelevant for those situations or overinterpreted as being relevant. It’s good that the authors use the word “hypothesize” on line 496. Similarly, the authors could be more cautious in the paragraph of lines 559-567.

12. Lines 583-593. Along the same lines, aren’t there NEC crystal structures from various herpesviruses that don’t crystallize into hexagonal lattices? The authors might incorporate those findings into their discussion here. See also the paragraph of lines 645-656.

13. Line 605. Insert “as” between acts and a.

14. Lines 629-636. How does the paper from the Heldwein lab suggesting that capsid protein induces pentamer formation fit into this? It seems relevant to the discussion.

Reviewer #3: Figure 1a – resolution of the image is very low?

Fig. 2 Bar label formatting needs to be fixed.

Fig 2b – shouldn't this be UL31-null/WTUL34 complementation?

Fig. 2d – Is the labeling and the legend correct for this panel? Is this virus UL-34+, UL31WT-neg and SUP-UL31+?

Fig. 2d - Why does the labeling indicate “Vector (no UL31)”? If this virus expresses UL34-WT and SUP-UL31, why is the vector control titer significantly lower than Panel 2C with the wt-UL34 complementing the UKL34-Null/SUP-UL31 virus?

The NEC-SUPUL31 crystal structure is at relatively low resolution (3.9Å). Could the authors speak to the accuracy of the coordinates at this resolution relative to the RMSD values that they calculate?

PLOS authors have the option to publish the peer review history of their article (what does this mean?). If published, this will include your full peer review and any attached files.

Reviewer #1: No

Reviewer #2: No

Reviewer #3: No

Figure Files:

Data Requirements:

Reproducibility:

References:

---

## [Editor Report · Decision Letter 1]

1 Jan 2024

Dear Dr. Heldwein,

We are pleased to inform you that your manuscript 'The universal suppressor mutation restores membrane budding defects in the HSV-1 nuclear egress complex by stabilizing the oligomeric lattice.' has been provisionally accepted for publication in PLOS Pathogens.

Best regards,

Paul M Lieberman

Academic Editor

PLOS Pathogens

Alison McBride

Section Editor

PLOS Pathogens

Kasturi Haldar

Editor-in-Chief

PLOS Pathogens

orcid.org/0000-0001-5065-158X

Michael Malim

Editor-in-Chief

PLOS Pathogens

orcid.org/0000-0002-7699-2064

The authors have addressed all of the reviewer comments.
---

## [Editor Report · Acceptance letter]

11 Jan 2024

Dear Dr. Heldwein,

We are delighted to inform you that your manuscript, "The universal suppressor mutation restores membrane budding defects in the HSV-1 nuclear egress complex by stabilizing the oligomeric lattice.," has been formally accepted for publication in PLOS Pathogens.

Best regards,

Michael Malim

Editor-in-Chief

PLOS Pathogens

orcid.org/0000-0002-7699-2064